# Human cytomegalovirus-encoded US9 targets MAVS and STING signaling to evade type I interferon immune responses

Hyun jin Choi[1], Areum Park[1], Sujin Kang[1], Eunhye Lee[1], Taeyun A. Lee[1], Eun A. Ra[1], Jiseon Lee[2], Sungwook Lee[2] & Boyoun Park[1]

Human cytomegalovirus (HCMV) has evolved sophisticated immune evasion mechanisms that target both the innate and adaptive immune responses. However, how HCMV encoded proteins are involved in this immune escape is not clear. Here, we show that HCMV glycoprotein US9 inhibits the IFN-β response by targeting the mitochondrial antiviral-signaling protein (MAVS) and stimulator of interferon genes (STING)-mediated signaling pathways. US9 accumulation in mitochondria attenuates the mitochondrial membrane potential, leading to promotion of MAVS leakage from the mitochondria. Furthermore, US9 disrupts STING oligomerization and STING–TBK1 association through competitive interaction. Intriguingly, US9 blocks interferon regulatory factor 3 (IRF3) nuclear translocation and its cytoplasmic domain is essential for inhibiting IRF3 activation. Mutant HCMV lacking US7-16 is impaired in antagonism of MAVS/STING-mediated IFN-β expression, an effect that is reversible by the introduction of US9. Our findings indicate that HCMV US9 is an antagonist of IFN signaling to persistently evade host innate antiviral responses.

[1] Department of Systems Biology, College of Life Science and Biotechnology, Yonsei University, Seoul 03722, South Korea. [2] Immunotherapeutics Branch, Research Institute, National Cancer Center, Goyang 10408, South Korea. Correspondence and requests for materials should be addressed to S.L. (email: swlee1905@ncc.re.kr) or to B.P. (email: bypark@yonsei.ac.kr)

Many multicellular species have pattern recognition receptors to detect intracellular viral nucleic acids and trigger host defense mechanisms, including the production of type I interferons (IFN)[1]. In particular, cytosolic or nuclear DNA sensors, such as a DExD/H-box helicase (DDX41), Z-DNA binding protein 1 (ZBP1), and gamma-interferon-inducible protein 16 (IFI16) are essential for sensing viral DNA[2–6]. These DNA receptors transduce signals via stimulator of interferon genes (STING), an endoplasmic reticulum (ER)-resident adaptor protein[7,8]. In addition, retinoic-inducible gene (RIG)-I-like receptors, which sense viral RNA molecules, interact with mitochondrial antiviral-signaling protein (MAVS), an adaptor protein localized to the mitochondrial outer membrane[9,10]. MAVS and STING function as scaffolds by recruiting and activating protein kinase TANK-binding kinase 1 (TBK1), which phosphorylates the transcription factor interferon regulatory factor 3 (IRF3), leading to stimulation of type I IFN production.

Many viruses have evolved mechanisms to evade the host immune system[11,12]. Previous studies suggest that several RNA viral proteins inhibit MAVS/STING-mediated immune responses[13–17]. For example, HCV NS4B protein interacts with STING and blocks its interactions with both MAVS and TBK1[18,19]. Likewise, DNA virus-encoded proteins, such as human papillomavirus E7 and adenovirus E1A, counteract STING signaling, leading to suppression of IFN-β production[20]. In particular, human cytomegalovirus (HCMV) encodes homologs of certain host cytokines, chemokines, and their receptors to mimic and evade a host innate immune attack[21,22]. Additionally, HCMV downregulates the expression or activation of factors involved in the IFN pathway and blocks the RIG-I and IFI16 receptors[23–27]. Despite such findings, the question of which HCMV-encoded glycoproteins target major mediators of the MAVS and STING pathways has yet to be answered.

HCMV infection increases the expression of proinflammatory cytokines or chemokines in the early stages, thereby facilitating virus dissemination through recruitment of HCMV-susceptible cells[28–31]. Moreover, many studies suggest that HCMV can suppress innate immune responses at late times of infection, leading to viral persistence within the host[25,32–34]. Consistent with these findings, the HCMV-encoded glycoprotein US9, which is barely detectable in early phases, has been detected 6–8 h after infection and has peak expression at 48 h[25]. Therefore, US9 may be involved in long-term HCMV persistence or survival in host cells; however, this hypothesis is yet to be investigated.

In this study, we identify the first HCMV glycoprotein US9 as the suppressor of MAVS and STING-mediated signaling to inhibit IFN-β production and antiviral responses during late stages of HCMV infection. Mitochondrial US9 inhibits IRF3 activation through MAVS leakage from the mitochondria. Within the ER, US9 has a distinct function in disrupting signaling along the STING–TBK1 axis, which results in inhibition of IRF3 nuclear translocation and IFN-β production. Deletion of the C-terminal region of US9 ablates its ability to dampen the MAVS- and STING-mediated IFN response, suggesting that the C-terminal domain of US9 is critical for its function. Consistent with in vitro data, HCMV infection shows that US9 is an important viral factor for promoting the reduction of mitochondrial MAVS expression and STING–TBK complexes, disrupting IRF3 nuclear translocation, and consequently inhibiting IFN-β production. Therefore, our study identifies an essential mechanism of HCMV glycoprotein US9 for evasion of the host antiviral response.

## Results

**US9 inhibits the MAVS and STING-mediated IFN-β responses.** The HCMV genome consists of a unique long (UL) and a unique short (US) regions, and each region is flanked by terminal repeats (TR$_L$/TR$_S$) and internal repeats (IR$_L$/IR$_S$). US region-encoded US9 protein contains an N-terminal signal sequence, a luminal domain, a transmembrane (TM) domain, and a cytoplasmic tail (CT) domain (Fig. 1a). A previous study reported that US9 traffics to the ER and mitochondria using its signal sequence and C-terminal mitochondrial localizing signal (MLS), respectively[35]. To understand the function of US9, we first confirmed its subcellular localization using a biochemical approach. We fractionated US9-expressing HEK293T cells and observed that US9 was present in the cytosolic/ER, but not in the nuclear fraction, and was further enriched in the mitochondria fraction (Fig. 1b and Supplementary Fig. 1a). We also performed an immunofluorescence assay (IFA) to examine various subcellular organelles in N-terminal hemagglutinin (HA)- and C-terminal green fluorescence protein (GFP)-tagged US9-expressing HEK293T cells. These experiments demonstrate that US9 was distributed throughout the ER and mitochondria, but dramatically enriched within the larger spherical foci around the perinuclear region (Fig. 1c). These patterns were not due to the GFP-tag because we also observed similar subcellular distributions of C-terminal HA-tagged US9 in HEK293T cells (Supplementary Fig. 1b). Intriguingly, US9-enriched larger foci were associated with altered ER morphology, compared with cells without apparent US9 expression (Fig. 1c, first panels). Furthermore, US9-enriched spherical structures correlated with a fragmented or accumulated staining pattern of mitochondrial marker, Tom20 (Fig. 1c, second panels). Other organelles that were surveyed, cytoskeleton or nucleus, exhibited no evident colocalization with US9 (Supplementary Fig. 1c).

Because many lines of evidence show that various modifications provide a strong driving force for the formation of intracellular foci, we hypothesized that US9-accumulated spherical foci correlate with its dimerization, oligomerization, or ubiquitination. HEK293T cells expressing both GFP- and HA-tagged US9 were lysed and immunoprecipitated with anti-GFP antibody, followed by immunoblot analysis with anti-HA antibody. Notably, our data clearly suggest that US9 undergoes dimerization (Fig. 1d). To further examine whether US9 forms oligomers, lysates from US9-expressing cells were separated using semi-denaturing detergent agarose gel electrophoresis (SDD-AGE), a method used to detect large SDS-resistant protein aggregates[36]. We found that US9 produced a smear of high-molecular weight oligomers (Fig. 1e). Moreover, immunoblot analysis with anti-ubiquitin antibody revealed the presence of ubiquitinated US9, which appeared as a large molecular weight smear in the top panel (Fig. 1f). Consistent with our previous result, IFA showed that US9 was clearly ubiquitinated (Supplementary Fig. 1d). Based on these results, we examined whether US9 ubiquitination is linked to its stability, since the ubiquitin–proteasome pathway is a key regulatory mechanism involved in protein turnover[37]. Pulse-chase labeling experiments revealed that US9 is highly long-lived, even up to 12 h (Supplementary Fig. 1e). Thus, we conclude that HCMV US9 undergoes dimerization, oligomerization, and ubiquitination, which may facilitate US9-enriched intracellular aggregates and lead to altered ER morphology and aberrant mitochondrial staining.

As previously noted, ER and mitochondria are important platforms for antiviral signaling[7–10,38]. The observation that US9-expressing cells displayed altered mitochondrial/ER staining led us to hypothesize that US9 may affect ER/mitochondrial antiviral signaling. To test this, we first examined the effect of US9 on IFN-β production in cells stimulated with salmon sperm DNA or synthetic double-stranded RNA (poly(I:C)), which activates the STING or MAVS signaling cascade, respectively. US9

significantly inhibited IFN-β mRNA production (Fig. 1g). To further clarify if US9 indeed affects STING- or MAVS-mediated IFN-β production, we observed IFN-β mRNA levels in the absence or presence of US9. Consistent with previous reports[9,39–41,46], ectopic expression of MAVS or STING potently activated the IFN-β mRNA production (Fig. 1h, i). Strikingly, co-expression of US9 with MAVS or STING inhibited signaling by either pathway (Fig. 1h, i, right graphs). Furthermore, MAVS- or STING-induced IFN-β protein secretion levels were inhibited in cells transfected with US9 (Fig. 1j). Thus, these observations suggest that ER- or mitochondria-targeted US9 blocks STING- or MAVS-mediated IFN-β production.

**US9 downregulates MAVS expression.** To understand the detailed molecular mechanisms underlying US9 function in the blockage of MAVS-mediated signaling, we investigated whether US9 affects MAVS expression patterns. In control cells, MAVS was distributed evenly. However, in US9-expressing cells, dispersed US9 overlapped with MAVS, while US9-enriched foci lacked MAVS expression (Fig. 2a, i–iii). Based on these findings, we further examined whether US9 downregulates MAVS expression. To test this, we assessed the endogenous expression level of MAVS in HEK293T and HFF cells expressing empty vector or US9 by immunoblot assay with anti-MAVS antibody. Indeed, poly(I:C)-stimulated MAVS expression was clearly

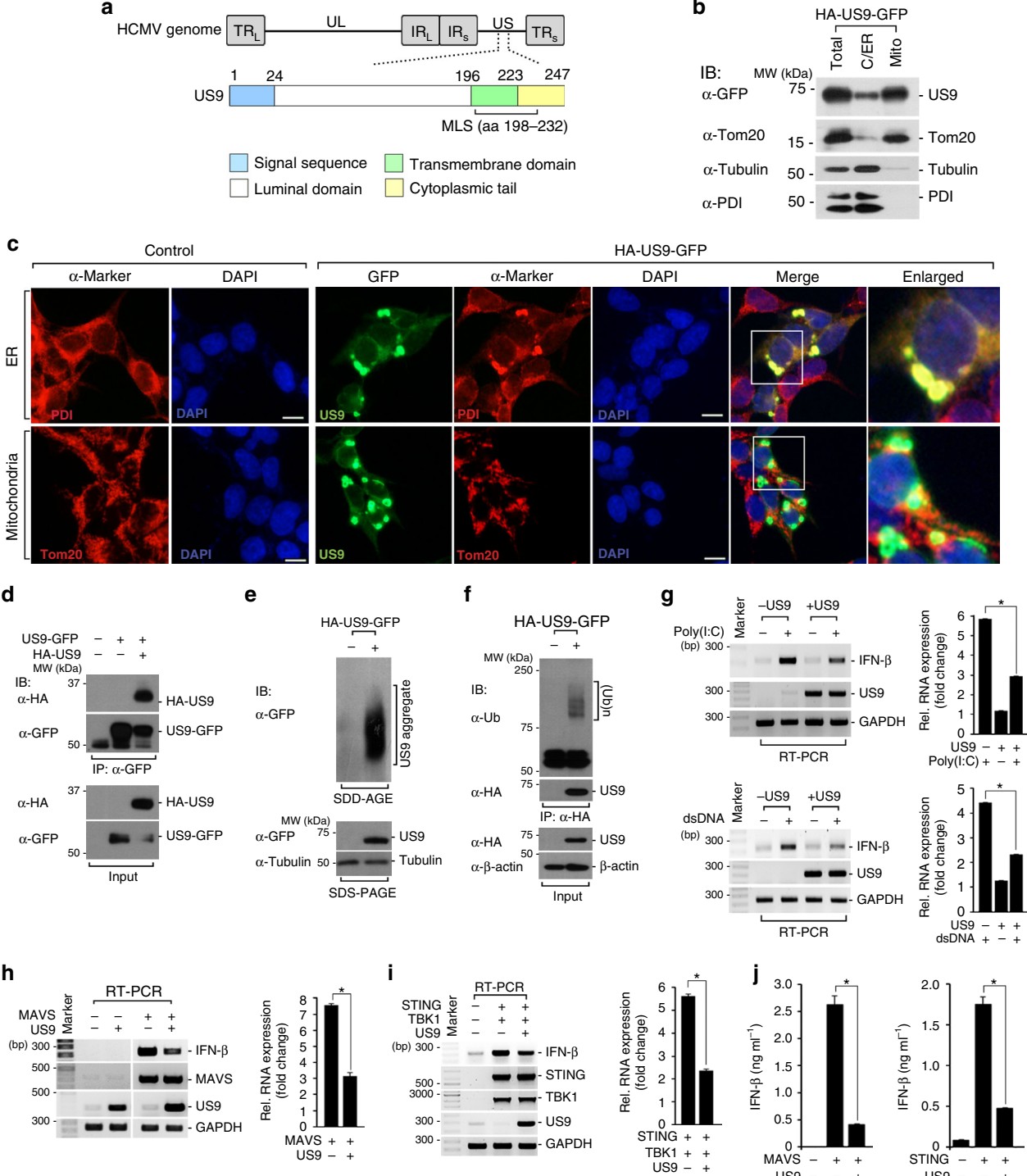

reduced by the presence of US9 (Supplementary Fig. 2a). Because overexpressing MAVS induces its mitochondrial aggregates and signaling pathway to produce IFN-$\beta$[9,39,47], we used this system to examine the effect of US9 on the downregulation of MAVS expression. Not surprisingly, US9-expressing cells showed a significant reduction in the abundance of Flag-tagged MAVS (MAVS-Flag) (Supplementary Fig. 2b). We further confirmed these findings via biochemical analysis using [35S]-methionine and cysteine-labeled cells. In the absence of US9, high expression of MAVS were detected after both the 0 and 4 h chases. In contrast, in US9-expressing cells, ~45% of labeled MAVS was reduced by the 4 h chase (Supplementary Fig. 2c, right graph). To address whether proteasomal or lysosomal degradation is involved in US9-mediated MAVS reduction, HEK293T cells were treated with either MG132, a proteasome inhibitor, or chloroquine, a lysosome inhibitor. Our data demonstrate that neither inhibitor affected US9-induced MAVS reduction (Supplementary Fig. 2d, e).

Next, we investigated whether US9-induced MAVS reduction involves direct interaction between these proteins. HEK293T cells were transfected with either MAVS-Flag alone or MAVS-Flag and US9-GFP, followed by immunoprecipitation of MAVS-Flag. Our results clearly suggest an interaction between US9 and MAVS (Supplementary Fig. 2f), which is consistent with our IFA and biochemical analysis with [35S]-labeling data showing colocalization of dispersed US9 and MAVS and detection of MAVS interaction with US9, respectively (Fig. 2a, iii and arrowhead of Supplementary Fig. 2c). To confirm these results, we assessed these interactions at endogenous levels. We clearly observed that endogenous MAVS interacted with US9 (Fig. 2b). Taken together, we conclude that US9 downregulates MAVS expression via a proteasome- or lysosome-independent pathway, and that the physical interaction between US9 and MAVS may be involved in US9-mediated attenuation of MAVS signaling.

**US9 induces MAVS leakage from the mitochondria.** Previous studies reported that loss of mitochondrial membrane potential ($\Delta\psi_m$) correlates with defects in the MAVS-mediated mitochondrial antiviral response[42,43]. Similarly, we found that Tom20 staining was reduced in US9-enriched larger foci of US9-expressing HEK293T cells (Fig. 1c). We thus hypothesized that mitochondrial US9 leads to defects in mitochondrial function. To test this hypothesis, we investigated the mitochondrial membrane potential in US9-expressing HeLa cells using the $\Delta\psi_m$-sensitive dye MitoTracker Orange. US9-expressing HeLa cells exhibited dramatically weak mitochondrial staining within the larger US9

foci regions, as well as fragmented mitochondria with an accumulation of punctate structures (Fig. 2c). Control cells, however, exhibited a uniform thread-like filamentous morphology. Furthermore, cells that co-expressed MAVS and US9 displayed more regions lacking MitoTracker staining and a higher degree of fragmented mitochondria (Fig. 2c). To address whether cell specificity is involved in US9-mediated $\Delta\psi_m$ disruption, we measured mitochondrial patterns in US9-expressing HEK293T cells. Similarly, US9-expressing HEK293T cells showed strong staining concomitant with dispersed US9 (arrowheads of Supplementary Fig. 2g). Interestingly, we observed significantly less mitochondrial staining in US9-enriched intracellular foci (arrowheads and arrows of Fig. 2c and Supplementary Fig. 2g). In contrast, the mitochondria in mock-transfected cells were stained uniformly. These findings suggest that mitochondrial localization of US9 leads to loss of $\Delta\psi_m$ and causes defects in mitochondrial dynamics. To determine whether US9 depolarizes the mitochondrial membrane, we examined the functional mitochondrial pool using MitoTracker Deep Red, a probe sensitive to $\Delta\psi_m$, and MitoTracker Green, a probe for mitochondrial membrane lipids independent of $\Delta\psi_m$. Markedly, co-expression of MAVS and US9 resulted in a higher increase in the mitochondrial damage index (MitoTracker Green positive and MitoTracker Deep Red negative) compared to control or MAVS expression alone in HEK293T cells (Fig. 2d).

A previous report showed that dynamic import of mitochondrial precursor proteins through the Tom complex channel at the mitochondrial outer membrane is required for maintaining mitochondrial functions, such as cellular signaling[44]. Therefore, we examined whether US9 affects Tom20 and Tom70, two components of the Tom complex. Co-immunoprecipitation (co-IP) experiments revealed that interaction of US9 with Tom20 was enhanced by the presence of MAVS (Fig. 2e). Notably, overexpression of US9 decreased the level of Tom70, which is important for the antiviral response by specifically interacting with MAVS[45] (Fig. 2f). We further hypothesized that US9 decreases Tom70 levels, thereby resulting in a decrease in the Tom20–Tom70 interaction on the outer mitochondrial membrane. Indeed, we clearly observed that US9 inhibited the Tom20–Tom70 interaction (Fig. 2g).

These results raised additional questions of whether US9-mediated abnormal mitochondrial integrity affects proper subcellular localization of MAVS, leading to inhibition of its signaling pathway and subsequent attenuation of IFN production. We thus examined the mitochondrial distribution of MAVS in US9-expressing cells. Interestingly, control cells expressing

**Fig. 1** ER- and mitochondria-targeted US9 inhibits the MAVS- and STING-mediated IFN-$\beta$ response. **a** Schematic representation of the HCMV genome and US9. MLS, mitochondrial localization signal. **b** US9 is localized to the ER and the mitochondria. Cytosolic/ER/mitochondrial fractions isolated from HA-US9-GFP-expressing HEK293T cells were immunoblotted with anti-GFP antibody. C/ER, cytosolic/ER fraction; Mito, mitochondrial fraction; Total, total cell lysate. Anti-Tom20, anti-Tubulin, and anti-PDI were used as the mitochondrial, cytosolic, and ER markers, respectively. **c** IFA of US9 subcellular localization. HEK293T cells expressing HA-US9-GFP were stained with an ER (anti-PDI) or mitochondrial marker (anti-Tom20). The nuclei were stained with DAPI. Scale bars, 10 μm. **d** US9 undergoes dimerization. Cell lysates from US9-GFP- and HA-US9-expressing HEK293T cells were immunoprecipitated with anti-GFP, and then immunoblotted with the indicated antibodies. **e** US9 forms high-molecular-weight oligomers. The cell extracts from HEK293T cells with or without HA-US9-GFP were resolved by SDD-AGE (top) and SDS-PAGE (bottom). **f** US9 undergoes ubiquitination. HEK293T cells expressing HA-US9-GFP were immunoprecipitated with anti-HA, and immunoblotted with anti-Ubiquitin. **g** US9 inhibits RNA- and DNA-induced IFN-$\beta$ induction. HeLa or HEK293T cells were transfected with poly(I:C) (20 μg ml$^{-1}$) or salmon sperm DNA (dsDNA) (2 μg ml$^{-1}$) for 12 h. IFN-$\beta$ mRNA levels were analyzed by RT-PCR. Right graph, quantification of IFN-$\beta$ mRNA expression levels was shown in the graph. *$P < 0.005$ (Student's $t$ test). **h** US9 impedes MAVS-mediated IFN-$\beta$ expression. HEK293T cells were transfected with the indicated plasmids and incubated for 21 h. Right graph, quantification of IFN-$\beta$ mRNA levels was shown in the graph. *$P < 0.005$ (Student's $t$ test). **i** US9 suppresses STING–TBK1-induced IFN-$\beta$ mRNA expression. HEK293T cells expressing STING-Myc and Flag-TBK1 were transfected mock or HA-US9-GFP. Right graph, quantification of IFN-$\beta$ mRNA levels was shown in the graph. *$P < 0.005$ (Student's $t$ test). **j** US9 blocks MAVS- and STING-induced IFN-$\beta$ production. HEK293T or HFF cells were transfected with MAVS or STING-Myc in the absence or presence of HA-US9-GFP, followed by transfection with salmon sperm DNA (1 μg ml$^{-1}$). After 36 h incubation, IFN-$\beta$ levels were analyzed by ELISA assay. *$P < 0.005$ (Student's $t$ test). Data are representative of three independent experiments and are presented as mean ± standard deviation (mean ± s.d.) in **g**, **h**, **i**, and **j**

MAVS-Flag alone exhibited exclusive MAVS localization within or adjacent to mitochondria, whereas US9-co-expressing cells showed abnormal localization of MAVS in regions that were distant from the mitochondria (Fig. 2h, arrowheads). These results led us to examine the possibility of MAVS leakage from the mitochondria as a result of US9 expression. In the presence of US9, MAVS was significantly reduced in the mitochondrial fraction and it partially leaked out into the cytoplasmic/ER fraction (Fig. 2i). Taken together, these data suggest that US9

induces MAVS leakage from mitochondria via disruption of the mitochondrial membrane potential and integrity, which inhibits MAVS-mediated IFN production.

**US9 disrupts STING–TBK1 signaling.** Because HCV NS4B is targeted to the ER and interacts with STING, leading to inhibition of STING-mediated IFN signaling[18,19,48], we hypothesized that ER-localized HCMV US9 negatively regulates STING–TBK1-

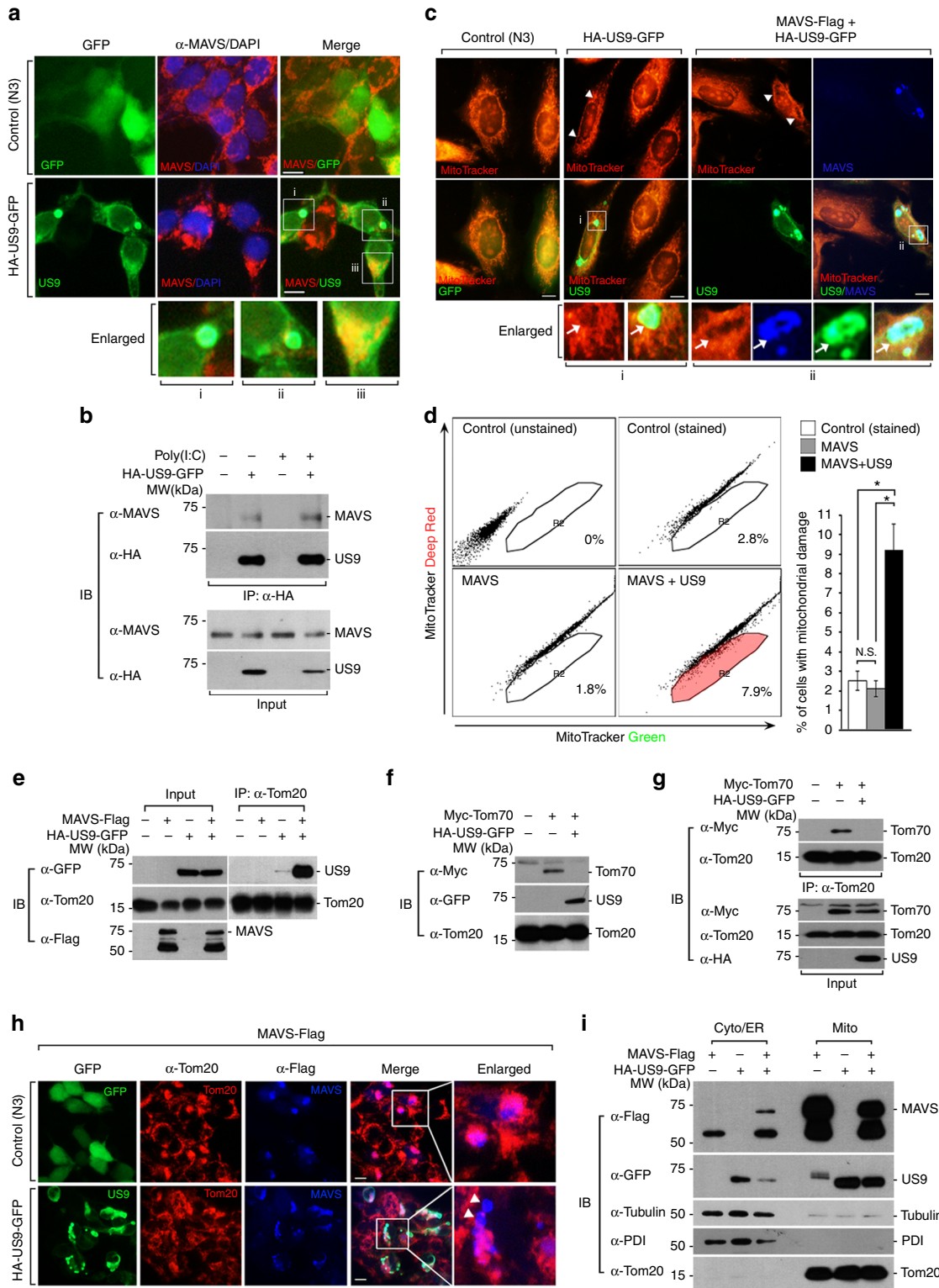

mediated signaling. To test this, we examined whether US9 colocalizes with STING or TBK1. Consistent with previous reports[8,49], we observed STING aggregates in STING-Myc-overexpressing cells (Fig. 3a). Notably, extensive colocalization of US9 with STING in aggregate structures was observed (Fig. 3a). To confirm this, we assessed the endogenous colocalization of STING with US9 in HFF cells that highly expressed endogenous STING (Fig. 3b, Supplementary Fig. 3a). Although the size of US9-GFP spherical foci in HFF cells was slightly smaller than that in US9-GFP-expressing HEK 293T cells, US9 clearly overlapped with endogenous STING aggregates and TBK1 in DNA-stimulated HFF cells (Fig. 3b, c). In fact, US9 strongly associated with both STING and TBK1 (Fig. 3d). Moreover, the association of STING and TBK1 was significantly disrupted by US9 compared with control cells (Fig. 3e, right graph). These results were not due to US9-mediated downregulation of STING or TBK1 levels since the expression level of these proteins did not differ between HFF or monocyte (U937) cells containing the US9 expression vector and an empty control vector (Supplementary Fig. 3b).

STING undergoes dimerization and polymerization to serve as a scaffold to recruit TBK1 and IRF3. Moreover, TBK1 promotes STING dimerization and subsequent oligomerization, which is critical for triggering type I IFN signaling[41,50,51]. We thus examined the effect of US9 on STING activation via dimerization and oligomerization. We found that US9 abolished STING dimerization and reduced high-molecular-weight STING aggregation, which was shown by the previously described SDD-AGE method (Fig. 3f). Taken together, these findings suggest that US9 inhibits STING/TBK1 complex formation and STING polymerization via its physical interaction with STING or TBK1.

**US9 interferes with IRF3 nuclear translocation**. We next explored whether US9-mediated inhibition of MAVS/STING signaling indeed targets the downstream transcription factor IRF3, which is critical for eliciting type I IFN expression to block viral replication[8,9]. Despite induction of IRF3 nuclear translocation by MAVS or STING overexpression, MAVS/STING-mediated IRF3 nuclear translocation was abrogated by US9 expression (Fig. 4a, b). We also counted the number of cells that exhibited nuclear IRF3 in empty vector (pEGFP-N3)- or US9-expressing cells. More than 90% of cells that expressed US9 exhibited severe disruption of IRF3 nuclear translocation compared to control cells (Fig. 4a, b). Interestingly, our finding that US9 colocalizes with cytosolic IRF3 raised the possibility of an interaction between US9 and IRF3. Indeed, co-IP analysis demonstrated that

US9 interacts with endogenous IRF3 (Fig. 4c). Because IRF3 phosphorylation is essential for its nuclear translocation, we also assessed the effect of US9 on IRF3 phosphorylation in response to MAVS or STING activation. Phosphorylated IRF3 levels were significantly decreased in US9-expressing cells compared with control cells transfected with the empty vector (Fig. 4d). This indicates that US9 facilitates IRF3 cytosolic sequestration through a physical interaction between US9 and IRF3, leading to dysfunctional nuclear translocation of IRF3. Taken together, these results demonstrate that US9 inhibits MAVS- and STING-induced IFN-β production via leakage of mitochondrial MAVS, blockade of the STING–TBK1 axis, and sequestration of cytosolic IRF3.

**The C-terminal domain of US9 is essential for its function**. Because the C-terminal domain of US9 resides in the cytoplasm, where many immune-related proteins are present for their role in immune functions, we speculated that its C-terminal domain might contribute to the suppression of MAVS- and STING-mediated IFN-β production. To test this idea, we generated a US9-deletion mutant lacking the C-terminal domain (US9ΔCT; Fig. 5a) and assessed whether this mutant undergoes proper folding by monitoring protein stability and fast-mobility in non-reducing conditions[52–55]. The US9ΔCT mutant was highly stable for chase times and exhibited a faster mobility than in non-reducing conditions. This result is similar to the pattern exhibited by full-length US9, supporting the idea that US9ΔCT maintains its proper folding (Supplementary Fig. 4a, b). Similar to wild-type US9, we observed US9ΔCT aggregates with altered ER morphology and fragmented and accumulated staining of Tom20 (Supplementary Fig. 4c). In addition, expression of the US9ΔCT-enriched foci dissipated the $\Delta\psi_m$ as shown by less staining by MitoTracker in US9ΔCT aggregate regions (Supplementary Fig. 4d). On the contrary to these phenomena, RT-PCR result intriguingly showed that US9ΔCT expression failed to inhibit MAVS-induced IFN-β activation, in contrast to wild-type US9 (Fig. 5b). We also observed that US9ΔCT restored mitochondrial MAVS levels (Fig. 5c). To demonstrate that US9ΔCT affects the MAVS-mediated IFN-β production, we further assessed IRF3 nuclear translocation in MAVS-overexpressing cells. As expected, IRF3 was clearly relocated to the nucleus in MAVS-Flag-expressing cells (Fig. 5d). US9ΔCT significantly restored MAVS-mediated IRF3 nuclear translocation compared with wild-type US9 (Fig. 5d). To confirm this result, we counted the number of cells with IRF3 nuclear translocation in US9- or US9ΔCT-expressing cells. Over 70% of the cells that expressed US9ΔCT

**Fig. 2** US9 reduces mitochondrial MAVS by disrupting mitochondrial membrane potential. **a** US9 colocalizes with endogenous MAVS. HEK293T cells transfected with HA-US9-GFP were visualized by staining with anti-MAVS antibody. The nuclei were stained with DAPI. Scale bars, 10 μm. **b** US9 interacts with endogenous MAVS. HEK293T cells were transfected with HA-US9-GFP followed by poly(I:C) (10 μg ml$^{-1}$) for 12 h. These cells were lysed and subjected to immunoprecipitation with anti-HA antibody prior to immunoblot analysis with the indicated antibodies. **c** US9 dissipates $\Delta\psi_m$ and alters mitochondrial structure. HeLa cells expressing control vector (pEGFP-N3), HA-US9-GFP, or both HA-US9-GFP and MAVS-Flag were incubated with MitoTracker Orange CMTMRos and immunostained with anti-Flag antibody, followed by AlexaFluor 350-conjugated antibody (blue). Boxed images are highlighted examples. Arrowheads or arrows indicate loss of $\Delta\psi_m$. Scale bars, 10 μm. **d** US9 increases damaged mitochondria with loss of $\Delta\psi_m$. HEK293T cells were transfected as indicated and were stained with MitoTracker Deep Red and MitoTracker Green for 30 min, and then FACS analysis was performed. *$P < 0.005$ (Student's $t$ test). **e** US9 associates with Tom20. Cell lysates were immunoprecipitated with anti-Tom20 antibody, and then subjected to immunoblot analysis using the indicated antibodies. **f** Tom70 protein level is reduced by US9. Cell lysates were subjected to immunoblot analysis with the indicated antibodies. **g** US9 disrupts Tom20–Tom70 complex formation. HEK293T cells co-expressing Myc-Tom70 and HA-US9-GFP were lysed, immunoprecipitated with anti-Tom20 antibody, and then analyzed by immunoblotting with the indicated antibodies. **h** MAVS localization is distorted by US9. HEK293T cells expressing MAVS-Flag with control vector (pEGFP-N3) or HA-US9-GFP were immunostained with anti-Tom20 and anti-Flag antibodies, followed by AlexaFluor 568- and AlexaFluor 350-conjugated antibodies, respectively. Arrowheads indicate distorted localization of MAVS into regions that were distant from the mitochondria. Scale bars, 10 μm. **i** US9 reduces mitochondrial MAVS and induces MAVS leakage into the cytosolic/ER fraction. HEK293T cells expressing MAVS-Flag in the absence or presence of HA-US9-GFP were separated into cytosolic/ER (Cyto/ER) and mitochondrial (Mito) fractions. Immunoblot analysis was performed using the indicated antibodies. Data are representative of three independent experiments and are presented as mean ± s.d. in **d**

restored IRF3 nuclear translocation, while wild-type US9-mediated IRF3 disruption was observed (Fig. 5d). Moreover, the interaction of US9ΔCT with IRF3 was significantly impaired compared to wild-type US9 (Fig. 5e). These results suggest that the C-terminal region of US9 is critical for downregulating mitochondrial MAVS levels and disrupting MAVS-induced IRF3 nuclear translocation, thereby suppressing MAVS-mediated IFN-β signaling.

Next, we investigated whether the C-terminal region of US9 also affects STING signaling. In contrast to wild-type US9,

US9ΔCT failed to inhibit STING-mediated IFN-β induction (Fig. 5f). To examine whether the C-terminal domain is required to abrogate STING–TBK1 interaction, we performed co-IP analysis. As previously noted, wild-type US9-expressing HEK293T cells showed a significant reduction in STING–TBK1 association (Fig. 3e). However, the same effect on the binding affinity of STING to TBK1 was observed in cells expressing US9ΔCT (Supplementary Fig. 4e). We sought to examine if the C-terminal domain of US9 may contribute to the STING dimerization. Interestingly, US9ΔCT restored STING

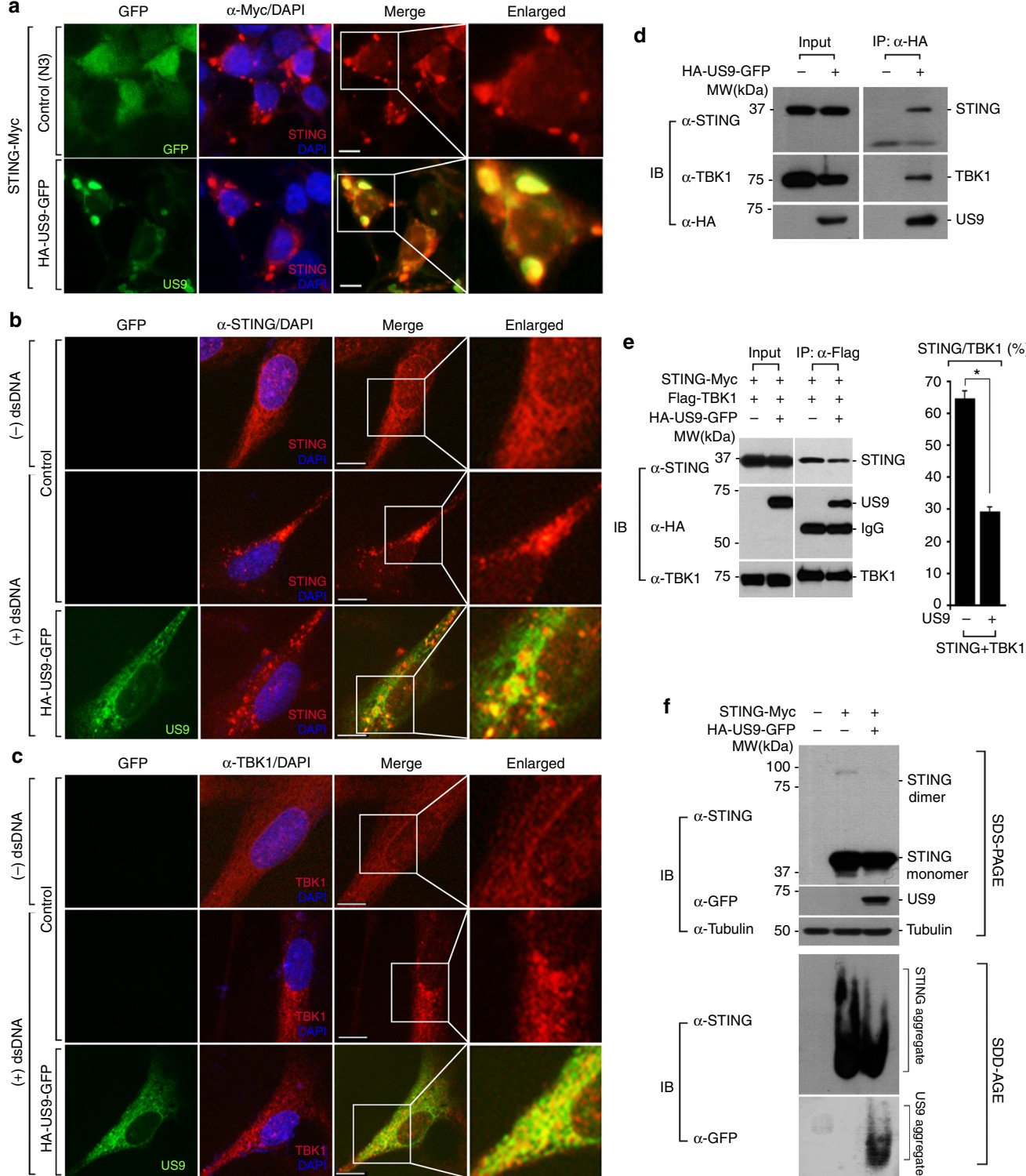

dimerization, while wild-type US9 significantly disrupted its dimerization (Fig. 5g). Additionally, IFA data clearly showed that US9ΔCT failed to disrupt STING-mediated IRF3 nuclear translocation (Fig. 5h). Taken together, these findings suggest that the C-terminal domain of US9 is essential for suppression of STING-mediated IRF3 nuclear translocation, disruption of STING dimerization, and subsequent inhibition of IFN-β production.

**HCMV US9 blocks IFN-β signaling in vivo.** To ascertain the physiological relevance of US9 in downregulating of IFN-β signaling, we explored US9 function in the context of HCMV infection. To test this, we first examined time-course of US9 expression levels in HCMV-infected cells. Consistent with a previous study[25], US9 expression increased steadily at the early time points, peaked at 2 days, and was sustainably expressed at later days during HCMV infection (Fig. 6a). To investigate whether US9 indeed affects IFN-β mRNA production, HFF cells were infected either with wild-type HCMV AD169 (HCMV WT), a HCMV deletion mutant lacking the US7–US16 region (HCMVΔUS7-16), nonessential for viral replication, or with a HCMVΔUS7-16 restored with US9 (HCMVΔUS7-16 + US9) (Fig. 6b). Because poly(I:C) stimulates IFN-β production through the MAVS pathway, when delivered into the cytosol by transfection, we observed a higher increase in IFN-β mRNA production (Fig. 6c, lane 2). In HCMV WT-infected HFF cells, IFN-β mRNA expression was significantly reduced, whereas in HCMVΔUS7-16, it was comparable to levels observed in poly(I:C)-stimulated uninfected cells (Fig. 6c, lanes 3 and 4). Markedly, HCMVΔUS7-16 + US9-infected cells showed a significant decrease in IFN-β mRNA expression, similar to that in HCMV WT-infected cells (Fig. 6c, lane 5). All HCMV viruses were successfully infected and robust expression of US9 was unambiguously detected in HCMV WT- and HCMVΔUS7-16 + US9-infected HFF cells, but not in HCMVΔUS7-16 (Supplementary Fig. 5, Fig. 6c). In particular, consistent with the observed results for HA-US9-GFP-expressing cells, HCMVΔUS7-16 + US9-infected HFF cells exhibited US9-enriched spherical foci (Fig. 6d). Moreover, IFA clearly showed that US9 led to altered ER morphology and aberrant mitochondrial patterns, and colocalized with endogenous MAVS, STING, and TBK1 in HCMVΔUS7-16 + US9-infected HFF cells, which are also consistently observed in HA-US9-GFP-expressing cells (Fig. 6d, e, f). Particularly, colocalization of US9 with MAVS, STING, and TBK1 in aggregate structures was clearly observed in HCMVΔUS7-16 + US9-infected HFF cells (Fig. 6e, f). To address whether US9 induces the leakage of mitochondrial MAVS in vivo, we purified mitochondrial fractionations and examined mitochondrial MAVS expression in US9-expressing cells. HCMVΔUS7-16 infection enhanced mitochondrial MAVS for activation, which was similar results observed in uninfected cells (Fig. 6g, lane 3). However, consistent with our in vitro data, mitochondrial MAVS expression was significantly reduced in HCMVΔUS7-16 + US9-infected HFF cells (Fig. 6g, lane 4).

To further investigate whether US9 negatively regulates STING–TBK1 axis under HCMV infection, we examined the STING–TBK1 interaction during HCMV infection. Endogenous STING–TBK1 interaction was markedly abrogated by HCMV WT even under DNA-stimulated conditions, but not HCMVΔUS7-16 (Fig. 6h). Intriguingly, introduction of US9 restored the inhibition of STING–TBK association in HCMVΔUS7-16 + US9-infected cells (Fig. 6h). We also examined the phosphorylated IRF3 nuclear translocation in nuclear fractions from WT- or mutant-HCMV-infected HFF cells. Consistent with in vitro results, IRF3 nuclear translocation was severely impaired in HCMV WT- and HCMVΔUS7-16 + US9-infected HFF cells, whereas HCMVΔUS7-16-infected cells induced translocation of IRF3 to the nucleus (Fig. 6i). Taken together, we conclude that HCMV US9 inhibits mitochondrial MAVS levels, STING dimerization and oligomerization, STING–TBK1 association, and IRF3 nuclear translocation, leading to inhibition of IFN-β expression in vivo (Fig. 7a).

## Discussion

In this study, we propose that US9 targets both MAVS and STING–TBK1 signaling pathways for inhibition of innate antiviral responses. Specifically, we found that US9 disrupts mitochondrial integrity and induces loss of mitochondrial membrane potential, thereby facilitating mitochondrial MAVS leakage and subsequently blocking MAVS signaling pathway. Moreover, we demonstrated that US9 inhibits the binding affinity of STING to TBK1 through competitive interaction with both STING and TBK and it blocks STING dimerization as well as polymerization that are important for STING activation and transduction of the signaling cascade, resulting in blockade of STING–TBK1-mediated IFN-β expression (Fig. 7b). In particular, US9 expression increased steadily at the early time points, maximized at 2 days after HCMV infection, when the inhibition of IFN-β signaling clearly occurred. Indeed, our findings in vitro and in vivo support that US9 facilitates MAVS degradation, disrupts the STING–TBK1 signaling axis, and impedes phosphorylated IRF3 into the nucleus during HCMV infection, resulting in inhibition of IFN-β expression. Overall, these results delineate a new role for US9 as a critical HCMV viral protein that negatively regulates host innate immune signaling.

Interestingly, accumulation of US9 at the mitochondria led to abrogation of the Tom20–Tom70 interaction by competitively binding to Tom20 and reducing the levels of Tom70 (Fig. 2). Several studies propose that Tom70 is essential for importing mitochondrial dynamin like GTPase optic atrophy 1 (OPA1) into mitochondria that maintains mitochondrial membrane potential and integrity[56,57]. Particularly, HCMV inhibits adenosine

**Fig. 3** US9 targets STING–TBK1 signaling axis to inhibit IFN-β signaling. **a** US9 colocalizes with STING as aggregates. HEK293T cells expressing STING-Myc were transfected with control vector (pEGFP-N3) or HA-US9-GFP. Cells were stained with anti-Myc antibody, followed by AlexaFluor 568-conjugated antibody. The nuclei were stained with DAPI. Scale bars, 10 μm. **b**, **c** US9 colocalizes with endogenous STING and TBK1. HFF cells stably expressing control vector or HA-US9-GFP were stimulated by transfecting with salmon sperm DNA (dsDNA) (1 μg ml⁻¹) for 12 h. Cells were stained with anti-STING (**b**) or anti-TBK1 (**c**) antibodies. The nuclei were stained with DAPI. Scale bars, 10 μm. **d** US9 interacts with endogenous STING and TBK1. HFF cells stably expressing control vector or HA-US9-GFP were stimulated by transfecting with salmon sperm DNA (dsDNA) (1 μg ml⁻¹) for 12 h. Cells were lysed and immunoprecipitated with anti-HA antibody, followed by immunoblotting with the indicated antibodies. **e** US9 disrupts the STING–TBK1 interaction. HEK293T cells expressing STING-Myc and Flag-TBK1 were transfected with or without HA-US9-GFP. For the immunoprecipitation assay, cells were lysed and the lysates were immunoprecipitated with anti-Flag antibody prior to immunoblot analysis with the indicated antibodies. The right bar graph represents the percentage of immunoprecipitated STING with TBK1. *$P < 0.005$ (Student's $t$ test). **f** US9 inhibits STING dimerization and aggregation. HEK293T cells expressing STING-Myc with or without HA-US9-GFP were lysed and cell lysates were subjected to SDS-PAGE analysis to determine STING dimerization (top) or SDD-AGE analysis to determine STING aggregation (bottom) following immunoblotting with the indicated antibodies. Data are representative of three independent experiments and are presented as mean ± s.d. in **e**

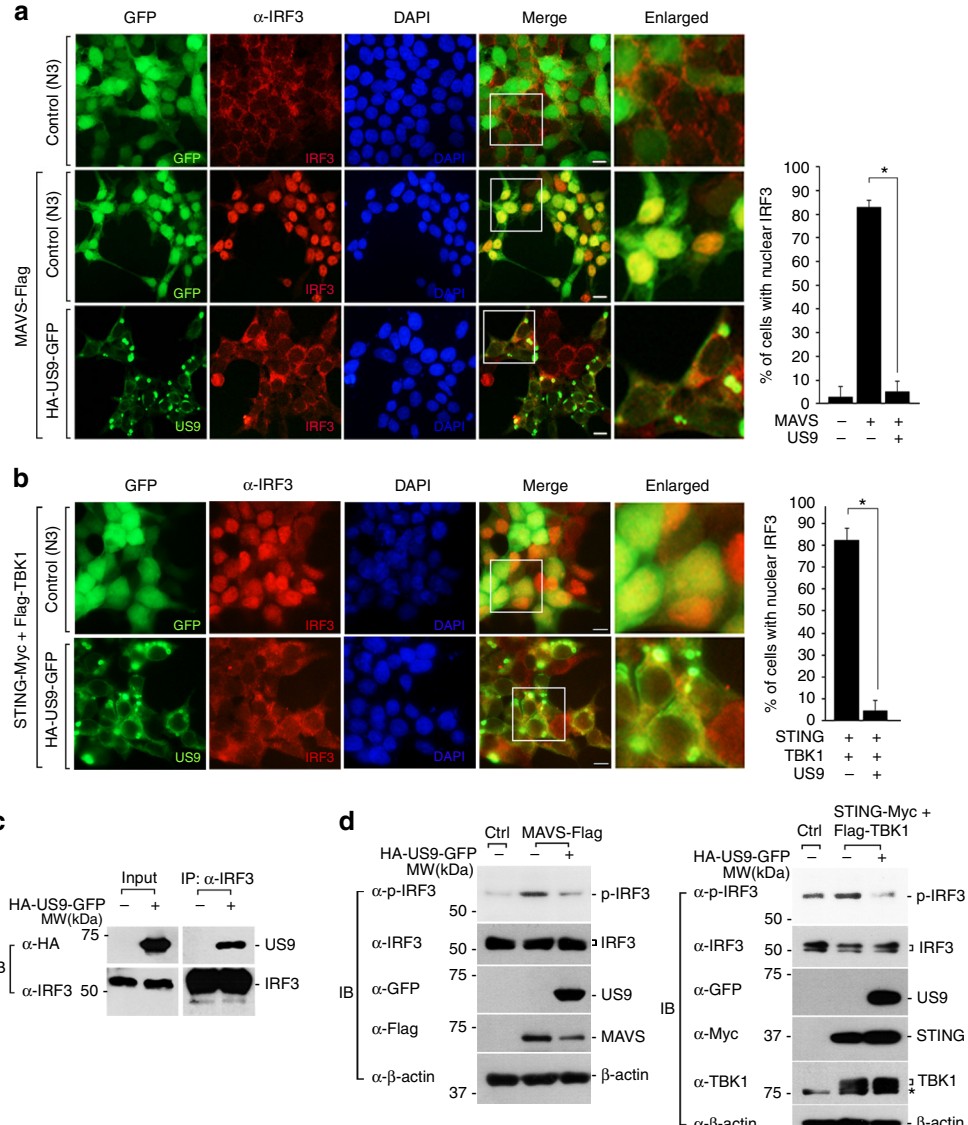

**Fig. 4** US9 impedes MAVS- and STING-mediated IRF3 nuclear translocation. **a** US9 blocks MAVS-induced IRF3 nuclear translocation. HEK293T cells were transfected with the indicated plasmids and immunostained with anti-IRF3 antibody, followed by AlexaFluor 568-conjugated antibody. The nuclei were stained with DAPI. Scale bars, 10 μm. Right graph, percentage of cells with nuclear IRF3. Scale bars, 10 μm. *$P < 0.005$ (Student's $t$ test). **b** US9 blocks STING–TBK1-induced IRF3 nuclear translocation. HEK293T cells expressing STING-Myc and Flag-TBK1 were transfected with HA-US9-GFP. IFA was performed by immunostaining with anti-IRF3 antibody. The nuclei were stained with DAPI. Right graph, percentage of cells with nuclear IRF3. Scale bars, 10 μm. *$P < 0.005$ (Student's $t$ test). **c** US9 interacts with IRF3. Cell lysates of HEK293T cells expressing mock or HA-US9-GFP were subjected to immunoprecipitation with anti-IRF3 antibody. Immunoprecipitates were resolved by SDS-PAGE and then subjected to immunoblotting with anti-HA and anti-IRF3 antibodies. **d** US9 inhibits MAVS- and STING/TBK1-induced IRF3 phosphorylation. HeLa cells were transfected with the indicated plasmids and total lysates were analyzed by immunoblotting with the indicated antibodies. Asterisk indicates endogenous TBK1. Data are representative of three independent experiments and are presented as mean ± s.d. in **a** and **b**

triphosphate (ATP) generation in the mitochondria, which disrupts the actin cytoskeleton and subsequently facilitates viral replication[58]. Because US9 also reduced mitochondrial integrity (Fig. 2), US9 may possibly incur severe mitochondrial damage at late phases of HCMV infection, thereby enhancing viral replication and inhibition of type I interferon. These strategies may serve two ends; namely, HCMV amplifies the functions of both virus dissemination and immune evasion, achieving latency throughout the lifespan of the infected host.

A recent study proposed a bicistronic MAVS transcript, which translates the full-length MAVS that promote IFN induction or the truncated MAVS that attenuates full-length MAVS-activated

response for homeostasis[59]. Paradoxically, we observed that US9 reduces levels of full-length MAVS as well as its truncated form of ~50 kDa that was easily detectable in immunoblot gels (Fig. 2i). The purpose of US9-mediated truncated MAVS reduction is not yet answered; however, because the truncated MAVS promotes cell death[59], it is interesting to speculate that US9-mediated reduction of truncated MAVS levels may provide benefits to protect from host cell death, enabling it to amplify viral particles during infection.

Functional domain study of US9 revealed that the C-terminal region of US9 is critical for its immune evasion function. Although the subcellular localization of the US9ΔCT mutant was

similar to that of wild-type US9, we showed a significant defect in the ability of the US9ΔCT mutant to inhibit both MAVS- and STING–TBK1-mediated IFN-β signaling. Particularly, in STING signaling, US9ΔCT could disrupt the STING–TBK1 interaction similarly to the wild-type US9; however, the US9ΔCT mutant failed to reduce STING dimerization or IRF3 nuclear transloca- tion. The importance of the C-terminal domain may stem from its exposure to the cytoplasm, which contains diverse signaling molecules. Whether other domains of US9 may also contribute to its downregulatory capability requires further studies.

Many RNA and DNA viruses have developed mechanisms to perturb the host IFN response. Based on these findings, we sug- gest that HCMV US9 targets the MAVS- and STING-signaling

factors to evade IFN signaling in the host cells. These evidences provide the first details for understanding the molecular mechanism of HCMV-encoded glycoprotein to evade from the host innate immune system. Knowledge of this mechanism will improve understanding of HCMV pathogenesis and assist in future development of therapeutic modalities to cure auto- immune diseases that are characterized by chronic production of cytokines, including type I IFN.

## Methods

**Antibodies and reagents**. Antibodies against anti-TOM20 (sc-11415), anti-IRF3 (sc-9082), anti-GFP (sc-9996), and anti-Ub (sc-8017) were purchased from Santa Cruz Biotechnology (Santa Cruz, CA, USA). Anti-Lamin A/C (#2032), anti-STING

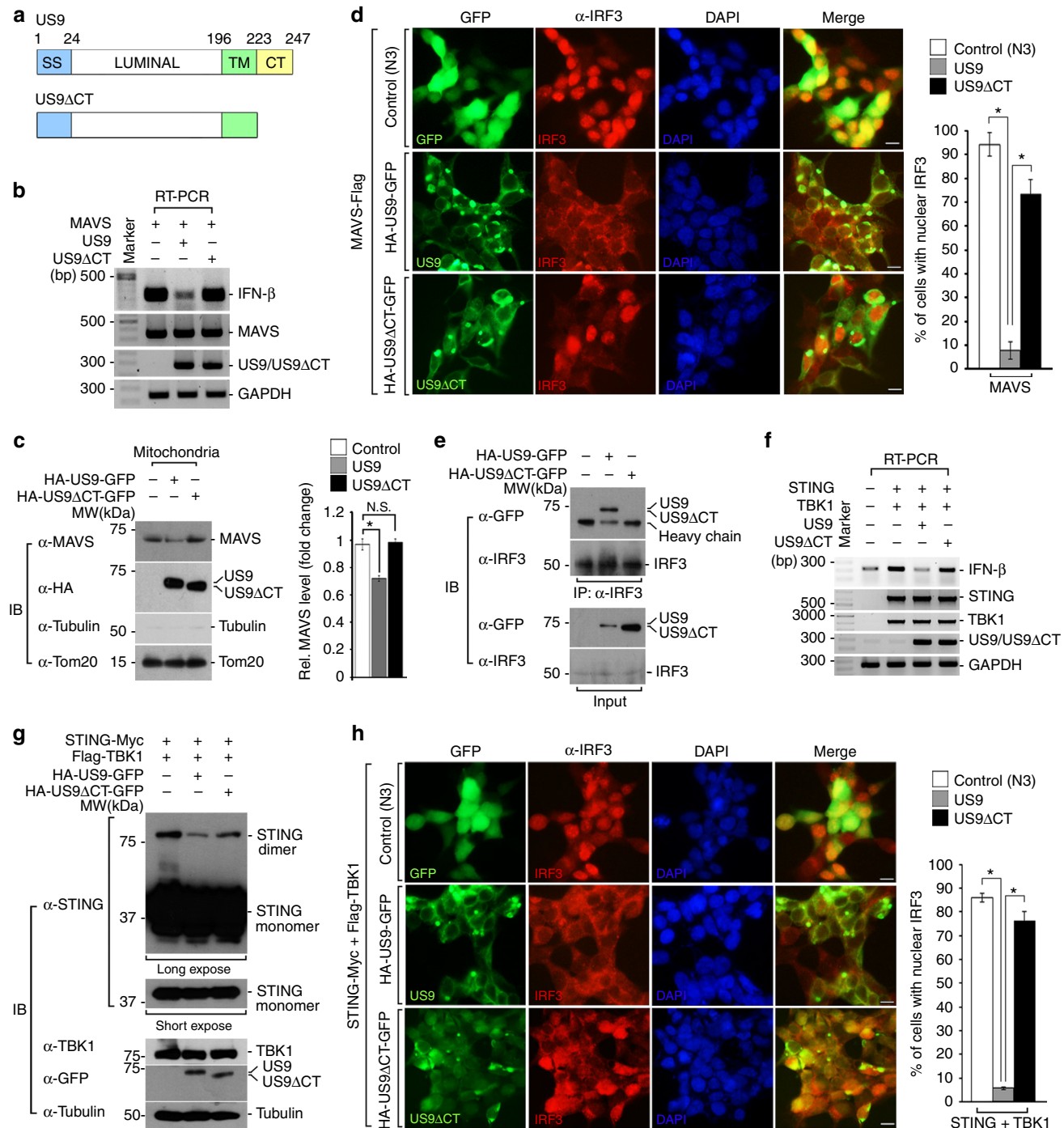

(#13647), anti-phospho-IRF3 (Ser396; #4947), and anti-Myc (#2276) antibodies were obtained from Cell Signaling Technology (Danvers, MA, USA). Anti-Tubulin (#G094), anti-FLAG (#G191), and anti-HA (#G036) antibodies were obtained from Applied Biological Materials Inc. (Richmond, Canada). Anti-NAK/TBK1 (#ab40676) antibodies were obtained from Abcam (Cambridge, UK). Anti-MAVS (#A300-782A) antibody was purchased from Bethyl Laboratories Inc. (Montgomery, TX, USA). Anti-protein disulfide isomerase (PDI; #ADI-SPA-891) was purchased from Enzo Life Sciences (Farmingdale, NY, USA). Anti-STING (MAB7169) obtained from R&D Systems (Minneapolis, MN, USA) was used for immunofluorescence assays. Anti-HA-Peroxidase, High Affinity (3F10) (# 12013819001) was purchased from Roche (Basel, Switzerland). The IE1 antibody (P63-27) was a gift from Dr Jun-Young Seo. MitoTracker Green FM (M7514) and MitoTracker Deep Red FM (M22426) were purchased from Invitrogen (Rockford, IL, USA). Poly(I:C), salmon sperm DNA, or Omicsfect (CP2101) were purchased from Sigma-Aldrich (St. Louis, MO) or Omics Bio (Taipei City, Taiwan), respectively. Digitonin (#300410) or Saponin (#47036) was obtained from Calbiochem/Millipore (Billerica, MA, USA) or Sigma (St. Louis, MO, USA). MitoTracker Orange CMTMRos (#M7510) and AlexaFluor 350-, AlexaFluor 488-, and AlexaFluor 568-conjugated antibodies were obtained from Life Technologies (Carlsbad, CA, USA).

**DNA constructs.** US9 fused with N-terminal HA tag was cloned into pcDNA 3.1 vector (Life Technologies) using the following primers: 5′-ATT GGT ACC TAC CCC TAC GAC GTC CCC GAC TAC GCC GGA ATT CTC GAG AAG GAG TCG CTC CGG TTG TC-3′ (KpnI; forward) and 5′-ATC TCT AGA TCA ATC GTC TTT AGC CTC TTC TTC C-3′ (XbaI; reverse). N-terminal HA tagged US9 was then subcloned into pEGFP-N3 vector (Clontech, Mountain View, CA, USA) using the following primers: 5′-ATG AGA TCT GCC ACC ATG GTC CCG TGC A-3′ (BglII; forward) and 5′-ATT GTC GAC ATC GTC TTT AGC CTC TTC TTC CG T-3′ (SalI; reverse). N-terminal HA and C-terminal GFP fused US9 construct was cloned into pMSCV vector (Clontech) by generating HA-US9-GFP insert from HA-US9-pEGFP-N3 plasmid (primers 5′-ATG AGA TCT GCC ACC ATG GTC CCG TGC A-3′ (BglII; forward) and 5′-ATT GTTTAAAC TTA CTT GTA CAG CTC GTC CAT-3′ (GFP-PmeI; reverse)), digesting pMSCV vector with BglII and HpaI, and ligating with the BglII and PmeI digested insert. US9 was cloned into pEGFP-N3 vector using the following primers: 5′-ATT AGA TCT GCC ACC ATG ATC CTG TGG TCC CCG-3′ (BglII; forward) and 5′-ATC TCT AGA ATC GTC TTT AGC CTC TTC TTC C-3′ (SalI; reverse). C-terminal HA fused US9 construct was cloned into pLHCX vector (Clontech) by generating US9 insert primers 5′-ATT AAGCTT GCC ACC ATG ATC CTG TGG TCC CCG-3′ (HindIII; forward) and 5′-ATT GTT AAC ATC GTC TTT AGC CTC TTC TTC CG T-3′ (HpaI; reverse), digesting pLHCX vector with HindIII and HpaI and ligating with the Hind III digested insert. For the C-terminal (CT) domain deleted construct (HA-US9ΔCT-pEGFP-N3), HA-tagged insert was generated from HA-US9-pEGFP-N3 using primers (5′-ATG AGA TCT GCC ACC ATG GTC CCG TGC A-3′ (BglII; forward) and 5′-ATT AAG CTT G GTG CCG ACC TCG CCA-3′ (HindIII; reverse) and ligated into pEGFP-N3. MAVS-Flag-pEF-BOS was purchased from Addgene (plasmid #27224). MAVS with no tag was subcloned into pcDNA3.1 using the following primers: 5′-ATT CTC GAG CCA CCA TGC CGT TTG CTG AAG ACA AGA CC-3′ (XhoI; forward) and 5′-ATT AAG CTT CTA GTG CAG ACG CCG CCG-3′ (HindIII; reverse). The human STING was cloned by PCR using the cDNA library of U937 cells using the following primers: 5′-ATT AAG CTT GCC ACC ATG CCC CAC TCC AGC CTG C-3′ (HindIII, forward) and 5′-ATT GGA TCC AGA GAA ATC CGT GCG GAG AGG G-3′ (BamHI; reverse). The human Flag-TBK1 was cloned by PCR using the cDNA

library mixture of HEK293, HEK293T, and HeLa cells using the following primers: 5′-ATT GGA TCC ATG CAG AGC-3′ (BamHI, forward) and 5′-ATT ATC GAT CTA AAG ACA GTC AAC GTT GCG-3′ (ClaI; reverse). Both were cloned into pLHCX vector (Clontech). Myc-Tom70 was cloned into pMSCV by generating insert with primer (5′-ATT CTC GAG ATG GCC GCC TCT AAA CCT G-3′ (XhoI; forward) and 5′-ATT GAA TTC TTA TCC TGT TGG TGG TTT TAA TCC G-3′ (EcoRI; reverse)), digesting Myc-tagged pMSCV vector with XhoI and EcoRI, and ligating with the insert. IFN-β promoter genes (−128 to + 21 bp) was prepared from human genomic DNA of HEK293T cells and cloned into the pGL3 basic vector (Promega, Madison, WI) with the following primers: 5′-ATT GAG CTC CTG AAT AGA GAG AGG ACC ATC TCA T-3′ (SacI; forward) and 5′-ATT CTC GAG AGG AGA GAA CAC TTG TTG GTC ATG TTG-3′ (XhoI; reverse). All constructs were verified by sequencing.

**Cell culture and transfection.** Human embryonic kidney (HEK) 293T cells (CRL-3216, American Type Culture Collection: ATCC, Manassas, VA), HEK293 cells (CRL-1573, ATCC), human foreskin fibroblast (HFF) cells (SCRC-1041, ATCC), and cervical cancer cell line HeLa (CCL-2, ATCC) were cultured in Dulbecco's Modified Eagle's Medium (DMEM) supplemented with 10% heat-inactivated fetal bovine serum (HyClone, Logan, UT, USA) and penicillin/streptomycin (HyClone). The monocytic cell line U937 (CRL-1593.2, ATCC) was cultured in RPMI containing 10% heat-inactivated fetal bovine serum and penicillin/streptomycin. All cells were grown at 37 °C in humidified air with 5% $CO_2$. Cells were transfected using Oimcsfect in serum-free and antibiotic-free DMEM for 20–36 h. For the maximal transfection efficiency of HFF cells, calculate the number of HFF cells plated to obtain 70% confluence and incubate the cells at 37 °C in $CO_2$ incubator for 23 h before transfection. All cell lines were tested for mycoplasma and were confirmed free of contamination.

**Retroviral transduction.** HEK293T cells were transfected with plasmids encoding VSV-G and Gag-Pol, as well as HA-US9-GFP. After 48 h post transfection, media containing viral particles were harvested and filtered with a 0.45 μm membrane. Cells were transduced with the virus by centrifugation at 2000 rpm for 45 min and incubated for 5 h. Transduced cells were incubated with fresh DMEM for 24–36 h and selected with puromycin.

**RT-PCR.** Total cellular RNA was isolated using an RNA prep kit (GeneAll, Seoul, South Korea), and 500 ng–1 μg of total RNA was reverse transcribed with oligo dT (18mer) primers using Moloney Murine Leukemia Virus (M-MLV) reverse transcriptase (Enzynomics, Daejeon, South Korea). Subsequently, PCR was performed with appropriate cDNA and primers, and the PCR products were visualized on ethidium bromide-stained agarose gels. All primer sequences used for RT-PCR are listed in Supplementary Table 1.

**Co-immunoprecipitation and immunoblotting.** Cells were lysed with 0.5% nonyl phenoxypolyethoxylethanol (NP-40, Sigma) in phosphate-buffered saline (PBS) or 1% digitonin with protease inhibitor cocktail (Roche) for 1 h at 4 °C. Cell lysates were incubated with primary antibodies overnight at 4 °C, and then protein G-Sepharose beads (Sigma) were added to the samples for 1–1.5 h at 4 °C. The beads were washed twice with 0.1% NP-40 in PBS. Proteins were eluted by boiling in 2× denaturing buffer (50 mM Tris-HCl, pH 6.8, 2% sodium dodecylsulfate [SDS], 5% β-mercaptoethanol). To completely elute and denature Tom20–Tom70 proteins, 2× denaturing buffer supplemented 15% β-mercaptoethanol was used. The samples were then separated by SDS-polyacrylamide gel electrophoresis (SDS-PAGE),

**Fig. 5** US9 cytoplasmic domain is important for inhibiting the IFN-β response. **a** Schematic representation of wild-type US9 and cytoplasmic tail (CT) domain truncation mutant (US9ΔCT). SS, signal sequence; LUMINAL, luminal domain; TM, transmembrane domain; CT, cytoplasmic tail. **b** The involvement of US9 C-terminal domain in the blockade of MAVS- induced IFN-β expression. After transfecting HEK293T cells with the indicated vectors, total RNA was isolated. Expression levels of IFN-β and other indicated genes were measured by RT-PCR analysis. **c** US9ΔCT restores mitochondrial MAVS levels. HEK293T cells were transfected with HA-US9-GFP or HA-US9ΔCT-GFP. Mitochondrial fraction was immunoblotted with the indicated antibodies. Anti-Tom20 and anti-Tubulin were used as the mitochondrial and cytosolic markers, respectively. *$P < 0.005$ (Student's $t$ test). **d** US9 CT domain is important for blocking MAVS-induced nuclear IRF3. HEK293T cells expressing MAVS-Flag were transfected with mock vector (pEFGP-N3), HA-US9-GFP, or HA-US9ΔCT-GFP. After 21 h, IFA was performed by immunostained with anti-IRF3 antibody. The nuclei were stained with DAPI. Scale bars, 10 μm. Right graph, percentage of cells with nuclear IRF3. *$P < 0.005$ (Student's $t$ test). **e** The C-terminal domain of US9 is essential to the interaction of US9 with IRF3. HEK293T cells expressing mock, HA-US9-GFP, or HA-US9ΔCT-GFP were lysed. The cell extracts were immunoprecipitated with anti-IRF3 and then subjected to immunoblot analysis with anti-GFP and anti-IRF3 antibodies. **f** The involvement of US9 C-terminal domain in the blockade of STING–TBK1-induced IFN-β expression. After transfecting HEK293T cells with the indicated vectors, total RNA was isolated. Expression levels of IFN-β and other indicated genes were measured by RT-PCR analysis. **g** The C-terminal region of US9 is important for inhibiting STING dimerization. STING-Myc and Flag-TBK1, together with HA-US9-GFP or HA-US9ΔCT-GFP, were expressed in HEK293T cells. Cells were lysed and then immunoblotted with the indicated antibodies. **h** The US9 C-terminal domain is critical for blocking STING–TBK1-mediated IRF3 nuclear translocation. HEK293T cells expressing STING-Myc and Flag-TBK1 in the presence of HA-US9-GFP or HA-US9ΔCT-GFP were analyzed by IFA with anti-IRF3 antibody. The nuclei were stained with DAPI. Scale bars, 10 μm. Right graph, percentage of cells with nuclear IRF3. *$P < 0.005$ (Student's $t$ test). Data are representative of three independent experiments and are presented as mean ± s.d. in **c**, **d** and **h**

transferred onto polyvinylidene fluoride membrane (Millipore, Billerica, MA, USA), blocked with 5% skim milk in PBS containing 0.1% Tween-20 (BioShop Canada Inc., Canada) (PBS-T) for 1 h, washed three times with PBS-T, and probed with appropriate primary antibodies overnight at 4 °C. Membranes were washed three times with PBS-T, and incubated with horseradish peroxidase-conjugated secondary antibodies for 1 h at room temperature. After washing three times with PBS-T, the immunoblots were visualized using enhanced chemiluminescence detection reagent (Advansta, Menlo Park, CA, USA). Uncropped images of western blots are shown in Supplementary Fig. 6.

**Pulse-chase analysis and immunoprecipitation.** Cells were starved for 1 h in medium lacking methionine and cysteine, labeled with 0.1 mCi ml$^{-1}$ [$^{35}$S] methionine/cysteine (PerkinElmer Life Sciences, Waltham, MA, USA) for 1 h, and chased in complete medium for the indicated times. After washing with PBS, the cells were lysed in 0.5% NP-40 in PBS for 1 h at 4 °C. Subsequent step for immunoprecipitation follows the method mentioned above. After separating the samples in SDS-PAGE, the gels were dried, exposed to BAS film, and analyzed using the phosphor imaging system (BAS-2500; Fuji Film, Tokyo, Japan).

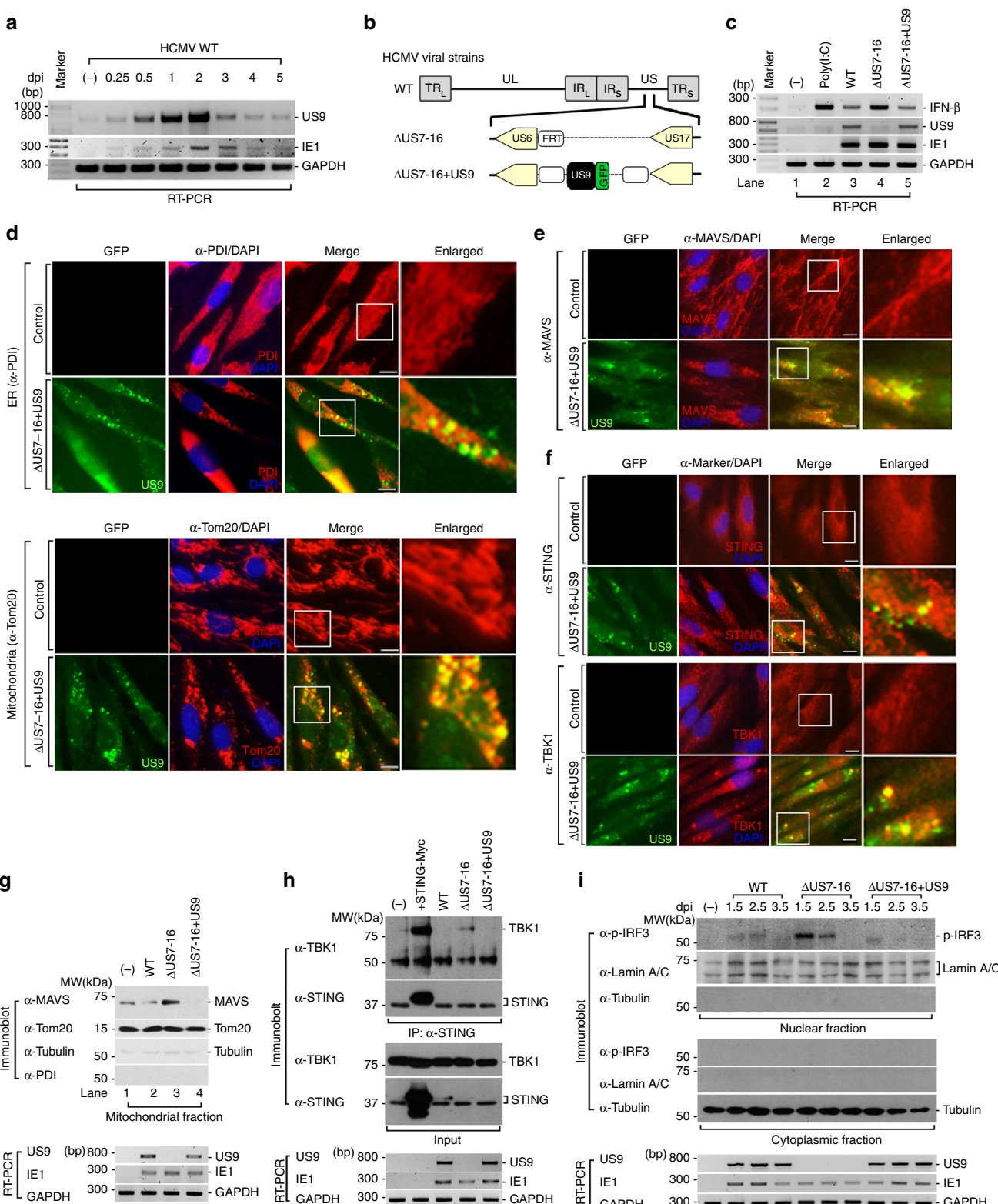

**Fig. 6** HCMV US9 inhibits MAVS- and STING-mediated IFN-β signaling. **a** US9 expression in HCMV WT-infected HFF cells. HFF cells were infected with HCMV WT at a multiplicity of infection (MOI) of 2. After incubating the infected cells for the indicated number of days post-infection (dpi), total RNA was isolated. The IE1 and GAPDH mRNA expressions were measured as virus infectivity and loading controls, respectively. **b** Schematic representation of the HCMV strains used. HCMV WT, wild-type HCMV AD169; HCMVΔUS7-16, HCMV deletion mutant lacking the US7–US16 region; HCMVΔUS7-16 + US9, HCMV mutant lacking all genes in the US7–US16 region restored with US9 expression. **c** HCMV US9 attenuates IFN-β expression. HFF cells were infected with HCMV WT, HCMVΔUS7-16, or HCMVΔUS7-16 + US9 at an MOI of 2 for 2 days (**c–h**). IFN-β mRNA levels were analyzed by RT-PCR. HFF cells transfected with poly(I:C) (20 μg ml⁻¹) for 12 h were used as a positive control. **d** ER- and mitochondria-targeted US9 correlates with altered ER/mitochondrial staining patterns. HCMVΔUS7-16+US9-infected HFF cells were then immunostained with an ER marker (anti-PDI) or mitochondrial marker (anti-Tom20). The nuclei were stained with DAPI. Scale bars, 10 μm. **e, f** HCMV US9 is colocalized with MAVS, STING, and TBK1. HCMVΔUS7-16+US9-infected HFF cells were immunostained with an anti-MAVS (**e**), anti-STING, or anti-TBK1 (**f**). The nuclei were stained with DAPI. Scale bars, 10 μm. **g** HCMV US9 downregulates mitochondrial MAVS expression levels in vivo. Mitochondrial fraction from HCMV-infected HFF cells was isolated and immunoblotted with the indicated antibodies. The presence of HCMV-derived US9 and IE1 expressions were analyzed by RT-PCR. **h** STING–TBK1 association is disrupted by HCMV US9. HFF cells (−) and HEK293T cells with salmon sperm DNA (2 μg ml⁻¹) for 12 h were used as negative and positive controls, respectively. US9 and IE1 expressions were analyzed by RT-PCR. **i** HCMV US9 blocks pIRF3 nuclear translocation in vivo. HFF cells infected with HCMV viruses at an MOI of 6 for 1.5, 2.5, or 3.5 days were subjected to nuclear fractionation, and then immunoblotted with anti-p-IRF3. Tubulin and Lamin A/C were analyzed as fraction loading controls. US9 and IE1 expressions were analyzed by RT-PCR. Data are representative of three independent experiments

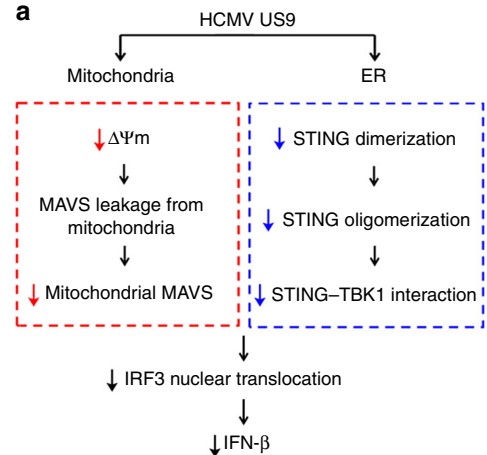

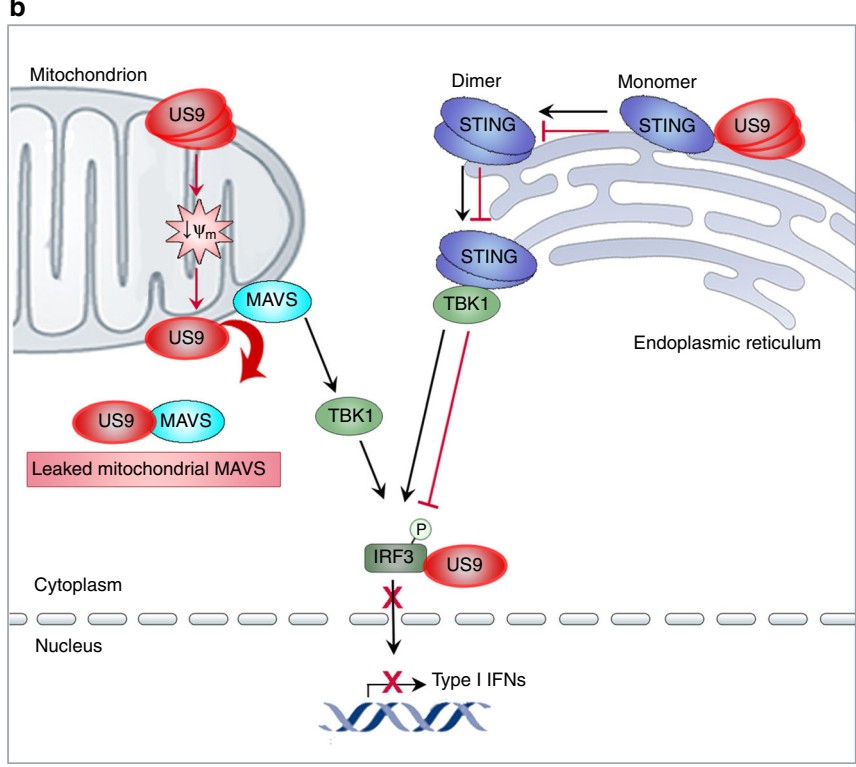

**Fig. 7** Proposed model of HCMV US9-mediated inhibition of IFN-β signaling. **a** Flowchart of US9-mediated negative regulation of IFN-β by targeting MAVS axis at the mitochondria and STING axis in the ER. **b** Schematic model of US9-mediated negative regulation of IFN-β by targeting MAVS and STING signaling pathways

**Flow cytometry**. To determine damaged mitochondria and mitochondria with mitochondrial membrane potential defect, cells were stained with 25 nM Mito-Tracker Green FM and MitoTracker Deep Red FM for 30 min at 37 °C. Cells were washed with PBS once and trypsinized by trypsin-EDTA, and resuspended in cold PBS with 1% FBS. Samples were immediately determined by FACSCalibur cytometer (BD Biosciences, San Jose, CA) and analyzed with CellQuest software (BD Biosciences).

**Mitochondria purification**. Mitochondria Isolation Kit for cultured cells (#89874, Thermo Scientific) was used to isolate cytosolic/ER and mitochondrial fractions. Briefly, $8 \times 10^6$ pelleted cells were resuspended with Mitochondria Isolation Reagent A, vortexed for 5 s, and then incubated on ice for 2 min. After adding Mitochondria Isolation Reagent B, samples were vortexed for 5 s, and incubated on ice for 5 min. Mitochondria Isolation Reagent C was then added and the tubes were inverted several times to mix. The supernatant obtained by centrifugation at 3200 rpm for 10 min at 4 °C was transferred to a fresh tube and centrifuged at 13,000 rpm for 15 min at 4 °C. The supernatant containing cytosolic/ER fraction was transferred to a new tube. Mitochondria Isolation Reagent C was added to the pellet, centrifuged at 13,000 rpm for 5 min, and the supernatant was discarded. The pellet containing the mitochondrial fraction was resuspended in 1× SDS loading buffer and analyzed by immunoblot analysis.

**Ubiquitination assay**. For ubiquitination assay, HEK293T cells transiently expressing indicated plasmids were incubated for 20–24 h. Cells were harvested and lysed in RIPA buffer (50 mM Tris-HCl (pH 8.0), 150 mM NaCl, 1% NP-40, 0.1% SDS, and 1 mM EDTA) containing protease inhibitor cocktail and 10 μM deubiquitinase inhibitor N-ethylmaleimide (NEM, Sigma). The cell lysate was immunoprecipitated with anti-HA antibody overnight at 4 °C and then protein G-Sepharose beads were added to the samples for 1–1.5 h at 4 °C. The beads were washed three times with RIPA buffer and proteins were eluted by boiling in 1× SDS loading buffer. Analysis of ubiquitination was performed by immunoblotting using anti-Ub antibody.

**Immunofluorescence assay**. For IFA, cells were fixed with 3.7% formaldehyde in PBS and permeabilized with 0.2% Triton X-100 in PBS before incubating with the appropriate primary antibody in 2% bovine serum albumin in PBS (PBA) for 1 h at room temperature or overnight at 4 °C. Bound antibody was visualized with Alexa Fluor 350-, Alexa Fluor 488-, or Alexa Fluor 568-conjugated secondary antibodies. The nuclei were stained with DAPI.

For immunofluorescence of HFF cells, cells were fixed with freshly prepared 3.7% formaldehyde in PBS and permeabilized with 0.1% saponin in 2% PBA for 20 min. Then primary antibody in 0.1% saponin in 2% PBA was incubated for 1 h at 37 °C followed by Alexa Fluor 568-conjugated secondary antibodies in 0.1% saponin in 2% PBA for 1 h at 37 °C. The nuclei were stained with DAPI.

**Cytosolic and nuclear protein fractionation**. Cells were lysed in cytosol extraction buffer (10 mM HEPES, pH 7.9, 10 mM KCl, 0.1 mM EDTA, 0.1 mM EGTA, 1 mM DTT, 0.5% NP-40) on ice for 15 min and centrifuged at 4000 rpm for 5 min. The supernatant was collected for cytosolic fraction. The pellet was lysed in hypertonic buffer (20 mM HEPES, pH 7.9, 0.4 M NaCl, 1 mM EDTA, 1 mM EGTA, 1 mM DTT) on ice for 20 min with vortexing every 5 min, and then centrifuged at 13,000 rpm for 20 min. The supernatant was collected for the nuclear fraction.

**Enzyme-linked immunosorbent assay**. HEK293T cells were seeded in six-well plates, transfected with MAVS with or without US9, and incubated for 36 h. HFF cells stably expressing mock or US9 expression were transfected with STING and stimulated with salmon sperm DNA (1 μg ml$^{-1}$) for 12 h. The cell culture supernatants were collected and human IFN-β level was analyzed using VeriKine human IFN-β ELISA kit (#41410-1, PBL Assay Science, NJ, USA) according to the manufacturer's instructions.

**Viruses**. HCMV wild-type AD169 and mutant AD169ΔUS7-16, in which US7-16 region is deleted and replaced with 48 bp FRT site. Recombinant HCMVΔUS7-16 + US9 used in the study was constructed utilizing a recombination strategy and the materials were provided by Dr Jun-Young Seo. Briefly, the Tet-on inducible US9-GFP fusion plasmid was inserted into mutant HCMV AD169ΔUS7-16 genome maintained in a bacterial artificial chromosome (BAC) using a recombination strategy. Escherichia coli (E. coli) SW105 strain, carrying BAC of mutant HCMV AD169ΔUS7-16 and ara-inducible Flpe recombinase gene, was transformed with pO6-SVT-entry plasmid containing US9-GFP. The ara-induced FLP recombinase induces site-specific recombination between the FRT sequences in the BAC and the insertion plasmid. To reconstitute the recombinant BAC containing US9-GFP into viruses, HFF cells were electroporated and were incubated until the recombinant HCMVΔUS7-16 + US9 viruses were generated. Virus stocks were prepared by infecting HFF cells with (MOI = 0.01) and incubating until 100% of cells showed cytopathic effects. Then cells were scraped and HCMV particle containing cell pellet and the supernatant were collected. The viral stocks were distributed in small aliquots, and stored at −80 °C. Virus stock aliquots were freshly thawed each time and not reused to avoid defective viral particles.

**HCMV infection**. HFF cells were plated in six-well plates and cultured in DMEM until cells were 70–80% confluent. HCMV strains were infected at a multiplicity of infection (MOI) as indicated. After incubating the infected cells for the indicated number of dpi, cells were washed and re-fed with fresh DMEM. Virus-infected cells were incubated for indicated time periods as indicated. To determine HCMV infectivity in all infection experiments, HCMV-infected HFF cells were stained with an anti-IE1 antibody and quantified by measuring viral infectivity.

**Statistical analysis**. All experiments were repeated at least three times with consistent results. Data are presented as mean ± s.d. (as noted in figure legends). Statistical differences between two means were evaluated with the two tailed unpaired Student's $t$ test. Differences with $P$-values below 0.05 were considered significant. Presented data were normally distributed and the variances were similar between the groups being statistically compared. No statistical method was used to predetermine sample sizes. Sample size was based on previous experience with experimental variability. No samples were excluded from the analysis. The experiments were not randomized. The investigators were not blinded to allocation during experiments.

**Data availability**. The authors declare that the data supporting the findings of this study are available within the article and its supplementary information files, or from the corresponding authors upon request.

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

## Acknowledgements

We thank Dr Jun-Young Seo (Severance Biomedical Science Institute, Yonsei University College of Medicine, Seoul, South Korea) for HCMV AD169, AD169ΔUS7-16, *E. coli* SW105 strain, pO6-SVT-entry plasmid, and HFF cells. This study was supported by grants from the Korean Health Technology R&D Project, Ministry of Health & Welfare, Republic of Korea (HI14C2542), and from Basic Science Research Program through the National Research Foundation of Korea (NRF) funded by the Ministry of Science, ICT and future planning (2015R1A2A1A15055053 and 2017R1E1A1A01074135). This work was also supported by the National Research Foundation of Korea (NRF) grant funded by the Korean Government (MSIP) (NRF-2016R1A5A1010764), and by the Strategic Initiative for Microbiomes in Agriculture and Food funded by Ministry of Agriculture, Food and Rural Affairs (916006-2). S.L. was supported by Basic Science Research Program through the National Research Foundation of Korea (NRF) funded by the Ministry of Education (2015R1D1A1A01060181), and a research grant from the National Cancer Center of Korea (NCC-1710210). H.J.C. was supported by NRF (National Research Foundation of Korea) grant funded by the Korean Government (NRF-2013- Global Ph. D. Fellowship Program). H.J.C., A.P., S.K., T.A.L., and E.A.R. were supported by Brain Korea (BK21) PLUS Program.

## Author contributions

H.J.C., A.P., S.K., E.L., T.A.L., E.A.R., and J.L. conducted experiments and analyzed data. H.J.C., S.L., and B.P. designed experiments and wrote the manuscript.

## Additional information

**Competing interests:** The authors declare no competing financial interests.

