## [Peer Review File · Nature Communications]

Reviewers' comments:

Reviewer #1 (Remarks to the Author):

The manuscript by Choi and colleagues examines the capacity of US9 regulatory protein of HCMV as a major antagonist of the innate antiviral responses, mediated by the cytosolic RNA and DNA sensor pathways controlled by MAVS and STING. The authors argue that a C-terminal domain of US9 is critical for the interaction with STING and MAVS and contributes to the disruption of STING dimer formation, as well as interactions with TBK and IRF3. Mitochondrial depolymerization mediated US9 localization to the mitochondria and interaction with Tom20. Many of the studies are well performed and reproducible, and support the concept that US9 is major antagonist of the antiviral response. A number of issues require further evaluation, comment, and experimentation.

Major issues:

A. Most of the experiments have been performed on transfected cells, studies on endogenous pathways are essential to confirm these results.

B. Furthermore, all the responses obtained with the US9 Δ CT mutant could be due to an alteration of the global folding rather than to the functionality of that particular domain. In order to exclude this eventuality, the authors should generate US9 mutants characterized by deletions of single amino acids indispensable for the C-term domain functions, but possessing identical conformation to the WT.

Other specific points for consideration.

Fig.1B They assess in the text that US9 distribution is restricted to the cytosolic compartment, but nuclear fraction is not shown

Fig.1C The authors described US9-transfected cells as having altered ER morphology and aberrant mitochondrial staining, but the figures shown are not representative of these conditions; in the lower panels, cells transfected (green) and not transfected with US9 share the same pattern of TOM20 staining

Fig.1D HA-US9 construct seems to have a molecular weight higher than 40kDa. Since US9 is a protein of 28KDa, is the HA tag they used is far longer than the classical 3xHA of 3kDa. What is the reason for this size discrepancy?

Fig. 1G, 6C, S2A: timing of poly I:C/dsDNA treatments are not specified

Fig. 2A, 2C: colocalization between US9 and endogenous vs transfected MAVS are discordant.

Fig S2E: the authors cited in the text arrowheads that are probably referred to fig. S2F

Fig. 2C, S4B: The authors describe regions negative for MitoTracker staining and positive for US9 aggregates as mitochondria with decreased membrane polarization. How can they exclude the possibility that those regions are not mitochondria?

Fig. 2D: How did the authors set the gate R2? The statement “co-expression of MAVS and US9 resulted in a higher increase in the mitochondrial damage index compared to expression of MAVS alone” does not consider that, according to the FACS analysis shown, MAVS transfected cells have less mitochondrial damage than control Hek293T.

Fig.2G: The authors performed CO-IP in denaturing conditions, hardly compatible with the detection of intact heterodimers in the final eluate.

Fig 2I: “overexpression of MAVS increased its enrichment in the mitochondrial fraction”; increase with respect to what? The authors never analyzed the subcellular localization of endogenous MAVS.

In immunofluorescent assays, do the authors consider control samples as cells transfected with a GFP plasmid (Fig. 2C, 2H, 4A, 5C, 5G, S1C) or not transfected at all (2A, 3A, 3B, 4B)?

Fig. 5C: Why do the authors comment that analysis of IRF3 nuclear translocation with: “US9-mediated IRF3 disruption was consistently observed”?

Fig. 5C, 5G: in resting conditions, the majority of control cells had IRF3 translocated in the nucleus. This point needs to be discussed, since it is inconsistent with much of the literature on IRF3.

Reviewer #2 (Remarks to the Author):

In this manuscript, Choi et al discover novel functions for the Human Cytomegalovirus US9 protein. Their data suggest that US9 targets both MAVS and STING signalling pathways, leading to inhibition of the IFN β response. As such, this paper is an important and novel addition to the field of HCMV biology and would be a candidate for publication in Nature Communications. However, I have some concerns that must be adequately addressed ('Major points') in addition to some more minor points, many of which can be addressed textually ('Minor points').

Major points

1. Much of the manuscript (the first 5 figures) relies on US9 that is overexpressed, either with a C-terminal GFP or C-terminal HA tag. The authors employ stably transduced cell lines to make the majority of conclusions in this paper, including multiple detailed studies with immunofluorescence. However, it is well known that overexpression, particularly of membrane proteins, can lead to artefactual subcellular localisation, bringing in to doubt a number of their conclusions. C-terminal tags can also modulate protein function/localisation (in this case they show that the C-terminus plays an important role in the interaction with MAVS and STING), and although they observe similar data with a US9 tagged with HA or GFP, both tags are at the C-terminus. In Figure 6, they show some nice data using a block-deletion (US7-16) virus with reconstituted US9-GFP, however do not use this virus to re-capitulate any of their immunofluorescent studies. It would seem that use of this virus could provide crucial confirmation of some of their observations – and this should be used to validate at least some of their IF studies in Figures 1-5.
2. What exactly are the large spherical perinuclear foci enriched in US9 (e.g. Figure 1C)? These only appear in some of the cells in the image shown (clearly could be present in another plane in the other cells – is this the case though)? In some cases they can apparently be quite massive, of comparable size to the nucleus (Figure S1B) – are these artefactual? How many are there per cell? Have similar structures previously been described? Confirmation that these are also seen with their DUS7-16+US9 virus is vital.
3. Their experiments with viral infection seem confounded by the use of a relatively modest moi. At moi 2, one would expect ~80% of cells to be infected, leaving 20% uninfected that may nevertheless produce IFN, particularly if exposed to defective viral particles (how was their viral prep

generated?) They should show the % infection they achieved with their wt, DUS7-16 and DUS7-16+US9 viral preps to confirm these were similar.

Minor points

1. Their assertion in the introduction that the temporal kinetics of US9 expression leads us to suppose that it 'may be an inhibitor that interferes with host innate immunity against HCMV' is an overstretch – by these criteria so would the majority of HCMV proteins. It is not clear from the manuscript how Choi et al actually identified US9 as a protein targeting the pathways they describe, however this statement should be modified.
2. My reading of Figure S2C is of a ~40% reduction of labelled MAVS by the 4h chase, not 85% (page 8 line 4)?
3. There is no arrowhead in figure S2E (page 8 line 15)
4. Experiments examining MAVS leakage from the mitochondria use ectopically expressed MAVS, which the authors acknowledge 'induced mitochondrial aggregates'. Is it possible to stain endogenous MAVS in IF rather than overexpressed? This point could be discussed in the text since the authors validate their finding with the DUS7-16+US9 virus. Similarly, can endogenous STING be detected for experiments in Figure 3?
5. How many replicates were used for e.g Figures 1H, 1I (they use a student's t-test so should detail this).
6. How efficient were their cellular transfections? I believe that HFF transfect very poorly – for Figure 1J this should be detailed.
7. Was Figure 2A stained with DAPI or a nuclear marker? This should be included.
8. Their finding that 'phosphorylated IRF3 levels were significantly decreased in US9-expressing cells compared with control cells transfected with the empty vector' (p12 line 13) might be due to decreased expression of MAVS induced by US9 – this should be discussed. Was the whole-cell level of MAVS reduced by US9, or just in certain subcellular structures?
9. Figures 4A-B and 5C examine nuclear localisation of IRF3 but no nuclear marker is shown – this should be corrected.
10. P15 line 9 should read 'similar TO results...'
11. The sentence 'In line with these reports, because US9 reduces the expression level of Tom70 and increased abnormal mitochondrial integrity (Figure 2), it is possible that US9 may occur a severe mitochondrial damage at later phases of HCMV infection, which enhances viral infectivity, along with inhibition of type I interferon.' doesn't make sense (p16-17)

12. Figure 5D apparently shows weaker binding of the C-terminal mutant of US9 to IRF3, however this is using two different constructs that may have different levels of overexpression. The blot with a-GFP input and a-IRF3 in the IP are both overexposed. A lower exposure of both parts should be shown and this blot should be quantified like most of the other blots in the manuscript.
13. The blot in Figure S4C isn't terribly convincing and is suggestive of variable transfer. Suggest repeat and quantify.

Comments from the reviewers that required a response are in bold italics, with each reply appearing in normal font just below the comment. In the replies, new material is emphasized in bold.

Reviewer #1 (Remarks to the Author):

The manuscript by Choi and colleagues examines the capacity of US9 regulatory protein of HCMV as a major antagonist of the innate antiviral responses, mediated by the cytosolic RNA and DNA sensor pathways controlled by MAVS and STING. The authors argue that a C-terminal domain of US9 is critical for the interaction with STING and MAVS and contributes to the disruption of STING dimer formation, as well as interactions with TBK and IRF3. Mitochondrial depolymerization mediated US9 localization to the mitochondria and interaction with Tom20. Many of the studies are well performed and reproducible, and support the concept that US9 is major antagonist of the antiviral response. A number of issues require further evaluation, comment, and experimentation.

Major issues:

A. Most of the experiments have been performed on transfected cells, studies on endogenous pathways are essential to confirm these results.

We agree with the reviewer's comments and have thus performed extensive co-IP/western blot/IFA analysis with endogenous proteins. Together with Figs. 1g, 4c, 5d, 6c, 6g~i, Suppl. Figs 1d, 2a, and 3, we have now provided new endogenous datasets in **New Figs. 2a, 2b, 3b, 3c, 3d, 5d, 6d~6h, Supplementary Figs. 4e, and 5** of the revised version.

B. Furthermore, all the responses obtained with the US9 Δ CT mutant could be due to an alteration of the global folding rather than to the functionality of that particular domain. In order to exclude this eventuality, the authors should generate US9 mutants characterized by deletions of single amino acids indispensable for the C-term domain functions, but possessing identical conformation to the WT.

The reviewer raises a valuable comment by asking whether the restorative effect of the US9 Δ CT mutant may be due to the global mis-folding of this mutant. If multiple amino acids in the C-terminal domain are involved in US9-mediated blockade of MAVS/STING signaling, it might possibly be difficult to determine the effect of deletions of single amino acids on its functional activity. Thus, to address the issue of proper global folding of the US9 Δ CT mutant, we performed two alternative experiments. Specifically, we conducted a pulse-chase assay with radioactive [³⁵S] methionine and cysteine, which is broadly used for monitoring the protein folding capacity of deletion or point mutants because misfolded proteins usually exhibit a significant reduction in protein stability because of proteasome/autophagy/chaperone-mediated degradation (*Ellgaard et al., Science, 1999; Olzmann et al., J. Cell. Biol. 2007; Kraft et al., Nat. Cell. Biol., 2010; Cuervo et al., Science, 2004; Yang et al., PNAS, 2011*). We clearly observed that the US9 Δ CT mutant was highly stable for chase times, similar to the pattern of full-length US9 (**New Supplementary Fig. 4a**). In addition, because US9 possesses two cysteine residues in the luminal domain (⁸¹Cys and ¹⁶⁴Cys) that are responsible for intra-disulfide bond formation, we performed a non-reducing SDS-PAGE analysis. This type of

analysis is widely used for measuring whether certain proteins are properly folded by the correct disulfide bond formation (*Marquardt et al., J. Cell Biol., 1992; Lee et al., Journal of Virology, 2000*). If proper disulfide bonds are formed, slightly faster mobility is observed because the existence of an intracellular disulfide bond maintains a more compact structure of the protein under non-reducing conditions. We observed that the US9 Δ CT mutant exhibits faster migrating during non-reducing SDS-PAGE analysis, which is similar to the migrating behavior of full-length US9 (**New Supplementary Fig. 4b**). These findings indicate we can exclude global folding problems as an issue with regard to the US9 Δ CT mutant. We included new datasets in **New Supplementary Figs. 4a, b** of the revised version. We also adequately included interpretations of these results and issues in the revised manuscript in response to the reviewer's comments.

Other specific points for consideration.

Fig.1B They assess in the text that US9 distribution is restricted to the cytosolic compartment, but nuclear fraction is not shown

We agree with the reviewer's point and have checked US9 distribution in both cytosol and nuclear fractions. Consistent with the previous results, we clearly observed US9 in the cytosolic fraction, but US9 proteins were barely detectable in the nuclear fraction (**New Supplementary Fig. 1a**).

Fig.1C The authors described US9-transfected cells as having altered ER morphology and aberrant mitochondrial staining, but the figures shown are not representative of these conditions; in the lower panels, cells transfected (green) and not transfected with US9 share the same pattern of TOM20 staining

We agree with the reviewer that US9-mediated aberrant ER and mitochondrial patterns were not clear in Fig. 1C. Thus, we repeated this experiment to achieve more convincing results. We provided more convincing, sharper images to clarify the altered ER morphology and dimly stained Tom20 patterns in US9-expressing cells (**New Fig. 1c**).

Fig.1D HA-US9 construct seems to have a molecular weight higher than 40kDa. Since US9 is a protein of 28KDa, is the HA tag they used is far longer than the classical 3xHA of 3kDa. What is the reason for this size discrepancy?

The reviewer has pointed out and questions the observation that HA-US9 exhibits a molecular weight higher than 40kDa. We used a vector system containing an amino-terminal affinity tag composed of an HA epitope followed by a tobacco etch virus (TEV) protease cleavage site and additional spacers attached to the US9 (HA-tev-US9), which is used as an affinity purification approach to identify US9-interacting partners (Lilley et al., *Nature*, 2004). We were remiss in not including detailed information of this construct in the Experimental Procedures.

To exclude the effect of the additional TEV site and spacers on US9 functional activity, we generated a new HA-US9 construct that only contains the N-terminally HA epitope and have performed a reciprocal IP/western blotting experiment with this construct (HA-US9). Consistent with the previous results with HA-tev-US9, we clearly showed that HA-US9 undergoes dimerization in a similar pattern as that of HA-tev-US9.

Because only HA-tev-US9 was used in the experiment shown in Fig. 1D throughout our manuscript, to avoid any confusion, we replaced the previous data using HA-tev-US9 with data for the new construct HA-US9 in **New Fig. 1d** of the revised manuscript. We also included detailed information of the HA-US9 construct in the figure legend and Experimental Procedures section of the revised manuscript.

Fig. 1G, 6C, S2A: timing of poly I:C/dsDNA treatments are not specified

We thank the reviewer for catching these errors and we added this information in the revised manuscript.

Fig. 2A, 2C: colocalization between US9 and endogenous vs transfected MAVS are discordant.

As previously described, we observed significantly less endogenous MAVS staining in US9-enriched intracellular foci due to MAVS leakage from the mitochondria and a decrease of MAVS expression levels as a result of US9-mediated aberrant mitochondrial integrity (Fig. 2 and Supplementary Figs. 2a~e). However, as the reviewer has pointed out, MAVS-Flag-expressing cells showed a considerable colocalization with US9 foci regions, possibly due to the very high overexpression of MAVS-Flag.

Nevertheless, we agree with the reviewer's comment. To clarify the colocalization between US9 and endogenous (vs transfected MAVS), we provided a more convincing and sharper image showing that US9 is clearly colocalized with endogenous and transfected MAVS (**New Figs. 2a, c**).

Fig S2E: the authors cited in the text arrowheads that are probably referred to fig. S2F

We thank the reviewer for identifying this error. We fixed the error in the revised manuscript.

Fig. 2C, S4B: The authors describe regions negative for MitoTracker staining and positive for US9 aggregates as mitochondria with decreased membrane polarization. How can they exclude the possibility that those regions are not mitochondria?

We provided image overlapping with MitoTracker staining and MAVS along with this response (please also see the additional figure at the end of this response). The regions negative for MitoTracker staining and positive for US9 aggregates clearly overlap, with the MAVS-enriched foci represented by blue color, and are known as a mitochondrial-localized protein. However, we agree with the reviewer's comment and, to avoid confusion, we provided sharper images that clearly show all staining colors (Mitotracker, MAVS, and US9) in **New Fig. 2c** of the revised manuscript.

Fig. 2D: How did the authors set the gate R2? The statement “co-expression of MAVS and US9 resulted in a higher increase in the mitochondrial damage index compared to expression of MAVS alone” does not consider that, according to the FACS analysis shown, MAVS transfected cells have less mitochondrial damage than control Hek293T.

The reviewer raises a valuable point in asking why MAVS transfected cells show less mitochondrial damage than control cells. We set the R2 region outline to select the negative population for mitochondrial damage in unstained control cells. However, we agree with the reviewer's comment that MAVS-expressing cells have shown less mitochondrial damage than control cells. To address this issue, we repeated this experiment three times independently and tried to determine whether ectopic expression of MAVS induces mitochondrial damage. However, no significant differences were observed in MAVS-expressing cells when compared to empty-vector-expressing cells (**New Fig. 2d**). We included a new graph along with FACS analysis datasets in the revised manuscript to display the new results.

Fig.2G: The authors performed CO-IP in denaturing conditions, hardly compatible with the detection of intact heterodimers in the final eluate.

We agree with the reviewer that there are not sufficient data to refer to a “Tom20-Tom70 complex”, because we previously detected this complex in denaturing conditions, possibly due to the incomplete reduction of Tom-20-Tom70 complex by low concentration of reducing agent, β -mercaptoethanol. Therefore, we conducted a co-IP/immunoblot experiment with 15% β -mercaptoethanol to determine whether US9 inhibits Tom20-Tom70 interaction. We clearly showed that US9 interrupts Tom20-Tom70 interaction through competitively binding of Tom20 and degrading Tom70 (**New Fig. 2e-g**). We provided new datasets and changed the text throughout our revised manuscript to better reflect the data presented.

Fig 2I: “overexpression of MAVS increased its enrichment in the mitochondrial fraction”; increase with respect to what? The authors never analyzed the subcellular localization of endogenous MAVS.

We thank the reviewer for identifying this discrepancy, and we amended this sentence in the revised manuscript.

In immunofluorescent assays, do the authors consider control samples as cells transfected with a GFP plasmid (Fig. 2C, 2H, 4A, 5C, 5G, S1C) or not transfected at all (2A, 3A, 3B, 4B)?

We agree with the reviewer's point that additional control samples were needed in our manuscript. Thus, we repeated most of the IFA assays with control samples to determine the specificity of GFP-tagged proteins. We provided new IFA datasets with controls in **New Figs. 2a, 3a, 4a, 4b, 5c, and 5g** of the revised manuscript.

Fig. 5C: Why do the authors comment that analysis of IRF3 nuclear translocation with: “US9-mediated IRF3 disruption was consistently observed”?

We thank the reviewer for catching this discrepancy. We amended this sentence in the revised

manuscript.

Fig. 5C, 5G: in resting conditions, the majority of control cells had IRF3 translocated in the nucleus. This point needs to be discussed, since it is inconsistent with much of the literature on IRF3.

As previously described in the early part of the text and figure legends, ectopic expression of MAVS-flag or STING-Myc itself is able to activate the IFN signaling pathway even without stimulation, leading to IRF3 nuclear translocation and IFN production (refs. 9, 39~41). Therefore, to show the effect of US9 on IRF3 nuclear translocation, we used MAVS-Flag or STING-Myc-expressing cells so that IRF3 was already transported to the nucleus. To avoid any confusion, we included additional descriptions in the text of the revised manuscript.

Comments from the reviewers that required a response are in bold italics, with each reply appearing in normal font just below the comment. In the replies, new material is emphasized in bold.

Reviewer #2 (Remarks to the Author):

In this manuscript, Choi et al discover novel functions for the Human Cytomegalovirus US9 protein. Their data suggest that US9 targets both MAVS and STING signalling pathways, leading to inhibition of the IFN β response. As such, this paper is an important and novel addition to the field of HCMV biology and would be a candidate for publication in Nature Communications. However, I have some concerns that must be adequately addressed ('Major points') in addition to some more minor points, many of which can be addressed textually ('Minor points').

Major points

1. Much of the manuscript (the first 5 figures) relies on US9 that is overexpressed, either with a C-terminal GFP or C-terminal HA tag. The authors employ stably transduced cell lines to make the majority of conclusions in this paper, including multiple detailed studies with immunofluorescence. However, it is well known that overexpression, particularly of membrane proteins, can lead to artefactual subcellular localisation, bringing in to doubt a number of their conclusions. C-terminal tags can also modulate protein function/localisation (in this case they show that the C-terminus plays an important role in the interaction with MAVS and STING), and although they observe similar data with a US9 tagged with HA or GFP, both tags are at the C-terminus. In Figure 6, they show some nice data using a block-deletion (US7-16) virus with reconstituted US9-GFP, however do not use this virus to re-capitulate any of their immunofluorescent studies. It would seem that use of this virus could provide crucial confirmation of some of their observations – and this should be used to validate at least some of their IF studies in Figures 1-5.

The reviewer raises an important point in asking about artefactual effects of an overexpression system and therefore requests additional IFA experiments with an HCMV deletion virus with reconstituted US9-GFP to confirm our results. We agree with the reviewer's comments and have performed extensive reciprocal IFA experiments in HFF cells infected with HCMV Δ US7-16+US9. Consistent with the IFA results of US9-transfected cells, we provided a new dataset and additional results with the virus system that show (a) a clear spherical perinuclear foci enriched US9 in HCMV Δ US7-16+US9-infected HFF cells, (b) US9 overlapped with MAVS and led to altered ER morphology and aberrant mitochondrial staining patterns, (c) clear colocalization of US9 with endogenous STING and TBK1, and (d) US9 enhances endogenous MAVS downregulation and abrogates STING-TBK1 interaction and IRF3 nuclear translocation. The new dataset with recombinant viruses is shown in **New Figs. 6d, 6e, and 6f** of the revised version, along with Figs. 6g, 6h, and 6i.

2. What exactly are the large spherical perinuclear foci enriched in US9 (e.g. Figure 1C)? These

only appear in some of the cells in the image shown (clearly could be present in another plane in the other cells – is this the case though)? In some cases they can apparently be quite massive, of comparable size to the nucleus (Figure S1B) – are these artefactual? How many are there per cell? Have similar structures previously been described? Confirmation that these are also seen with their DUS7-16+US9 virus is vital.

The reviewer raises a valuable point by requesting confirmational experiments with HCMVΔUS7-16+US9 virus to determine whether these large spherical perinuclear foci of US9 are observed in HCMVΔUS7-16+US9-infected cells. In HCMVΔUS7-16+US9-infected HFF cells, we clearly observed US9-enriched spherical foci similar to patterns shown in US9-GFP-expressing cells (**New Figs. 6d, 6e, and 6f**). The size of US9 spherical foci in HCMVΔUS7-16+US9-infected cells were slightly smaller than that in US9-GFP-expressing cells, possibly due to the very high overexpression of US9-GFP. As previously noted in the manuscript, because US9 undergoes dimerization and forms oligomers, US9-enriched foci are easily detectable like visible spherical foci enriched MAVS or STING oligomers are detectable in RNA- or DNA-stimulated cells, respectively.

3. Their experiments with viral infection seem confounded by the use of a relatively modest moi. At moi 2, one would expect ~80% of cells to be infected, leaving 20% uninfected that may nevertheless produce IFN, particularly if exposed to defective viral particles (how was their viral prep generated?) They should show the % infection they achieved with their wt, DUS7-16 and DUS7-16+US9 viral preps to confirm these were similar.

We agree with reviewer's comment. To determine HCMV infectivity in all infection experiments, HCMV-infected HFF cells were stained with an anti-IE1 antibody and quantified by measuring viral infectivity. The viral infectivity for wild-type or mutant HCMV viruses was approximately > 93% in HFF cells (**New Supplementary Fig. 5**). We included detailed information of the percent infection in the experiments with wild-type or mutant HCMV-infected HFF cells in the revised version.

Minor points

1. Their assertion in the introduction that the temporal kinetics of US9 expression leads us to suppose that it ‘may be an inhibitor that interferes with host innate immunity against HCMV’ is an overstretch – by these criteria so would the majority of HCMV proteins. It is not clear from the manuscript how Choi et al actually identified US9 as a protein targeting the pathways they describe, however this statement should be modified.

We agree with reviewer's comment and have changed the text throughout our revised manuscript to reflect the reviewer's points.

2. My reading of Figure S2C is of a ~40% reduction of labelled MAVS by the 4h chase, not 85% (page 8 line 4)?

We thank the reviewer for identifying mislabels and have fixed these errors in the revised manuscript.

3. There is no arrowhead in figure S2E (page 8 line 15)

We are grateful to the reviewer for catching this error. We amended the description to include the correct infectious doses in the results, experimental procedures, and appropriate figure legends in the revised manuscript.

4. Experiments examining MAVS leakage from the mitochondria use ectopically expressed MAVS, which the authors acknowledge ‘induced mitochondrial aggregates’. Is it possible to stain endogenous MAVS in IF rather than overexpressed? This point could be discussed in the text since the authors validate their finding with the DUS7-16+US9 virus. Similarly, can endogenous STING be detected for experiments in Figure 3?

The reviewer raises an important point. As described in our previous response, we observed significantly less endogenous MAVS staining in US9-enriched intracellular foci due to MAVS leakage from the mitochondria and a decrease of MAVS expression levels as a result of US9-mediated aberrant mitochondrial integrity (Fig. 2 and Supplementary Fig. 2a~e). However, as the reviewer has pointed out, MAVS-Flag-expressing cells showed a considerable colocalization with US9 foci regions, possibly due to the very high overexpression of MAVS-flag. To clarify the colocalization between US9 and endogenous (vs transfected MAVS), we provided a more convincing and sharper image showing that US9 is clearly colocalized with endogenous and transfected MAVS (**New Figs. 2a, c**).

In addition, we also performed extensive co-IP/western blot/IFA analyses with endogenous MAVS/STING/TBK proteins. We now provided new datasets in **New Figs. 2b, 3b, 3c, 3d, and 5d** of the revised version of the manuscript. As noted previously, HCMVΔUS7-16+US9-infected cells showed US9-mediated aberrant mitochondria/ER morphology and US9-STING/MAVS/TBK1 colocalization (**New Figs. 6d, 6e, and 6f**). We included new datasets and changed the text to reflect the reviewer's points in the revised manuscript.

5. How many replicates were used for e.g Figures 1H, 1I (they use a student's t-test so should detail this).

We repeated the RT-PCR experiment at least three times independently and all bands were quantified and normalized to GAPDH. As the reviewer suggested, we included detailed information of statistical analysis for all bar graphs in the figure legends of the revised manuscript.

6. How efficient were their cellular transfections? I believe that HFF transfect very poorly – for Figure 1J this should be detailed.

Although HFF cells are not good for transfection, as the reviewer points out, we tried to optimize experimental conditions to achieve maximal transfection efficiency (>80%; please see the additional figure **A** at the end of this response, next page). We also performed western blot analysis checking the expression level of transfected STING-Myc in HFF cells (please see the additional figure **B** at the end of this response). Therefore, we described detailed information for HFF transfection in the Experimental Procedures section of the revised manuscript.

7. Was Figure 2A stained with DAPI or a nuclear marker? This should be included.

We agree with the reviewer's comments and performed a reciprocal IFA experiment with nuclear marker staining. We provided new datasets with DAPI in the revised version of the manuscript.

8. Their finding that 'phosphorylated IRF3 levels were significantly decreased in US9-expressing cells compared with control cells transfected with the empty vector' (p12 line 13) might be due to decreased expression of MAVS induced by US9 – this should be discussed. Was the whole-cell level of MAVS reduced by US9, or just in certain subcellular structures?

As the reviewer pointed out, US9 facilitates MAVS leakage from the mitochondria and a significant decrease in its stability and expression levels in whole-cell lysates (Fig. 2 and Supplementary Fig. 2). Therefore, we revised the manuscript to reflect the reviewer's points.

9. Figures 4A-B and 5C examine nuclear localisation of IRF3 but no nuclear marker is shown – this should be corrected.

We agree with the reviewer's comments and have repeated the IFA experiments with DAPI staining. We provided new datasets in the revised version of the manuscript.

10. P15 line 9 should read 'similar TO results...'

We thank the reviewer for identifying errors and have amended the errors in the revised manuscript.

11. The sentence 'In line with these reports, because US9 reduces the expression level of Tom70 and increased abnormal mitochondrial integrity (Figure 2), it is possible that US9 may occur a severe mitochondrial damage at later phases of HCMV infection, which enhances viral infectivity, along with inhibition of type I interferon.' doesn't make sense (p16-17)

We thank the reviewer for identifying this discrepancy and have revised the text throughout the revised manuscript to reflect the reviewer's points.

12. Figure 5D apparently shows weaker binding of the C-terminal mutant of US9 to IRF3, however this is using two different constructs that may have different levels of overexpression. The blot with α -GFP input and α -IRF3 in the IP are both overexposed. A lower exposure of both parts should be shown and this blot should be quantified like most of the other blots in the manuscript.

We agree with reviewer's comment and have repeated this experiment to showing a more convincing result. We provided new datasets in the revised manuscript.

13. The blot in Figure S4C isn't terribly convincing and is suggestive of variable transfer. Suggest repeat and quantify.

We agree with the reviewer's comments and have performed reciprocal western blot experiments four times independently. We clearly observed that US9 Δ CT restores mitochondrial MAVS levels. As the reviewer suggested, the MAVS levels were quantified in the graph. We provided new datasets in **New Fig. 5c** of the revised manuscript.

REVIEWERS' COMMENTS:

Reviewer #1 has no further comments

Reviewer #2 (Remarks to the Author):

Choi et al have now submitted a detailed rebuttal with a number of comprehensive additional experiments. As such they have satisfied all of my initial concerns and I would recommend that the manuscript be published in it's revised format.